# Sketching Algorithms for Sparse Dictionary Learning: PTAS and Turnstile Streaming

**Gregory Dexter**
Department of Computer Science
Purdue University
gdexter@purdue.edu

**Petros Drineas**
Department of Computer Science
Purdue University
pdrineas@purdue.edu

**David P. Woodruff**
Computer Science Department
Carnegie Mellon University
dwoodruf@cs.cmu.edu

**Taisuke Yasuda**
Computer Science Department
Carnegie Mellon University
taisukey@cs.cmu.edu

## Abstract

Sketching algorithms have recently proven to be a powerful approach both for designing low-space streaming algorithms as well as fast polynomial time approximation schemes (PTAS). In this work, we develop new techniques to extend the applicability of sketching-based approaches to the *sparse dictionary learning* and the *Euclidean $k$-means clustering* problems. In particular, we initiate the study of the challenging setting where the dictionary/clustering *assignment* for each of the $n$ input points must be output, which has surprisingly received little attention in prior work. On the fast algorithms front, we obtain a new approach for designing PTAS's for the $k$-means clustering problem, which generalizes to the first PTAS for the sparse dictionary learning problem. On the streaming algorithms front, we obtain new upper bounds and lower bounds for dictionary learning and $k$-means clustering. In particular, given a design matrix $\mathbf{A} \in \mathbb{R}^{n \times d}$ in a turnstile stream, we show an $\tilde{O}(nr/\epsilon^2 + dk/\epsilon)$ space upper bound for $r$-sparse dictionary learning of size $k$, an $\tilde{O}(n/\epsilon^2 + dk/\epsilon)$ space upper bound for $k$-means clustering, as well as an $\tilde{O}(n)$ space upper bound for $k$-means clustering on random order row insertion streams with a natural "bounded sensitivity" assumption. On the lower bounds side, we obtain a general $\tilde{\Omega}(n/\epsilon + dk/\epsilon)$ lower bound for $k$-means clustering, as well as an $\tilde{\Omega}(n/\epsilon^2)$ lower bound for algorithms which can estimate the cost of a single fixed set of candidate centers.

## 1 Introduction

A classic idea in machine learning and signal processing for efficiently handling large datasets is to approximate them by simpler or more structured surrogate datasets. Many methods in this direction have long been considered, including low rank approximation, which approximates a given dataset by one that lies on a low-dimensional subspace, $k$-means clustering, which approximates a given dataset by at most $k$ distinct points, and sparse dictionary learning Olshausen and Field [1997], which approximates a given dataset by linear combinations of elements of a small dictionary of size $k$ with $r$-sparse coefficient vectors (i.e., a vector with at most $r$ nonzero entries). We focus on the latter two problems in this work:

**Definition 1.1** ($r$-sparse dictionary learning). *Let $\{a^i\}_{i=1}^n \subseteq \mathbb{R}^d$ be a set of $n$ vectors in $d$ dimensions, and let $\mathbf{A} \in \mathbb{R}^{n \times d}$ be the matrix with the $i$th row set to $a^i$. Then for a matrix $\mathbf{X} \in \mathbb{R}^{n \times k}$ with*

37th Conference on Neural Information Processing Systems (NeurIPS 2023).

*r-sparse rows and a dictionary* $\mathbf{D} \in \mathbb{R}^{k \times d}$, *we define the dictionary learning cost to be*

$$\mathrm{cost}(\mathbf{X}, \mathbf{D}) \coloneqq \|\mathbf{X}\mathbf{D} - \mathbf{A}\|_F^2$$

*In the $r$-sparse dictionary learning problem, we seek to minimize* $\mathrm{cost}(\mathbf{X}, \mathbf{D})$ *over all* $\mathbf{X} \in \mathcal{X}$ *and* $\mathbf{D} \in \mathbb{R}^{k \times d}$, *where* $\mathcal{X}$ *denotes the set of all $n \times k$ matrices with $r$-sparse rows.*

**Definition 1.2** (Euclidean $k$-means clustering)**.** *Let* $\{a^i\}_{i=1}^n \subseteq \mathbb{R}^d$ *be a set of $n$ vectors in $d$ dimensions, and let* $\mathbf{A} \in \mathbb{R}^{n \times d}$ *be the matrix with the $i$th row set to $a^i$. Then, for a matrix* $\mathbf{X} \in \mathbb{R}^{n \times k}$ *with standard basis vectors in its rows and a set of centers* $\mathbf{C} \in \mathbb{R}^{k \times d}$, *we define the $k$-means clustering cost to be*

$$\mathrm{cost}(\mathbf{X}, \mathbf{C}) \coloneqq \|\mathbf{X}\mathbf{C} - \mathbf{A}\|_F^2.$$

*In the $k$-means clustering problem, we seek to minimize* $\mathrm{cost}(\mathbf{X}, \mathbf{C})$ *over all* $\mathbf{X} \in \mathcal{X}$ *and* $\mathbf{C} \in \mathbb{R}^{k \times d}$, *where* $\mathcal{X}$ *denotes the set of all $n \times k$ matrices with standard basis vectors as rows.*

While dictionary learning and clustering have found extraordinary success in various applications in practice, they are known to be computationally difficult problems to solve [Mahajan et al., 2012, Natarajan, 1995], and thus there has been intense focus on developing approximation algorithms and heuristics for these problems, such as those based on greedy methods [Lloyd, 1982, Das and Kempe, 2011] or convex relaxations [Donoho and Elad, 2003, Fuchs, 2004, Cohen-Addad et al., 2022a].

In this work, we study algorithms for sparse dictionary learning and $k$-means clustering in two distinct settings via a unified set of techniques based on *sketching*. Sketching [Woodruff, 2014b], broadly speaking, refers to techniques for compressing large matrices by linear maps, and includes methods such as oblivious sketching and nonuniform sampling. Classically, sketching has been applied to design low-memory algorithms in the *streaming setting*, when the input is presented to the algorithm as a sequence of updates. More recently, sketching has been shown to be invaluable for designing fast algorithms as well. In particular, there has been a line of work which shows how sketching techniques can be applied to obtain *polynomial time approximation schemes* (PTAS) for a variety of NP-hard problems ranging from clustering [Feldman et al., 2007] to weighted low rank approximation [Razenshteyn et al., 2016] to tensor decompositions [Song et al., 2019]. We study such sketching-based algorithms for sparse dictionary learning and Euclidean $k$-means clustering, both in the offline setting where we obtain the first PTAS for sparse dictionary learning, as well as in the turnstile streaming and other streaming models. In particular, in the streaming setting, we initiate the study of solving these problems in the setting where the algorithm must output the assignment of the points to the dictionary/clustering, which has received surprisingly little attention in prior work.

## 1.1 Our contributions

### 1.1.1 PTAS for dictionary learning and clustering

We start with a discussion of our results on designing fast PTAS's. Our main contribution that we highlight from this section is the *first PTAS for sparse dictionary learning*, which also gives a new and simple approach towards designing a PTAS for $k$-means clustering.

A typical approach for designing PTAS's for shape fitting problems such as dictionary learning and clustering is to first find a smaller instance whose solution approximates the original instance, and then to solve the smaller instance using any algorithm, where even an inefficient algorithm will be tractable due to the smaller size of the instance. A representative work which takes such an approach for the $k$-means clustering problem is that of Feldman et al. [2007], which uses *coresets* to implement the first step of finding a smaller instance. Here, coresets for $k$-means clustering are a weighted subset of the original data points such that the cost of any candidate set of centers approximates the cost when applied to the original dataset. Furthermore, the size of this coreset can be taken to be $\mathrm{poly}(k/\epsilon)$, and thus solving for an optimal set of centers on this subset of points can be done in time independent of the number of points $n$. Due to this natural approach, there has been a long line of work on obtaining smaller coresets for $k$-means clustering [Feldman and Langberg, 2011, Braverman et al., 2016, Bachem et al., 2018, Cohen-Addad et al., 2021, 2022b,c].

On the other hand, for the sparse dictionary learning problem, similar results are strikingly lacking. The only previous work we are aware of is a coreset construction for the sparse dictionary learning problem due to Feldman et al. [2013]. However, the construction of the coreset in this work requires an algorithm for computing an approximately optimal dictionary, which prevents its use in designing

fast PTAS's to solve the dictionary learning problem in the first place. To address this problem, we first show that a completely different coreset technique due to Tukan et al. [2022] for the projective clustering problem can in fact be applied to the sparse dictionary learning problem. Notably, this technique uses John ellipsoids to construct coresets rather than using a nearly optimal solution to the dictionary learning problem, and thus avoids computing approximately optimal dictionaries. In turn, this allows us to obtain the first PTAS for the dictionary learning problem. Our argument additionally combines this coreset construction with a sparsity-counting technique together with polynomial system solvers Renegar [1992a,b] to efficiently solve a smaller version of the original problem. Our techniques also yield a new PTAS for $k$-means clustering, which is arguably simpler than prior approaches such as the algorithm of Feldman et al. [2007]. We give a full discussion of our results and techniques for our PTAS for sparse dictionary learning in Section 2.

### 1.1.2  Dictionary learning and clustering on streams

As our next contribution, we study algorithms for dictionary learning and clustering in turnstile streams and other related models of streaming. In the turnstile streaming model, the input undergoes arbitrary entrywise insertions and deletions:

**Definition 1.3** (Turnstile stream). *We say that an input matrix* $\mathbf{A} \in \mathbb{R}^{n \times d}$ *is presented in a* turnstile stream *if* $\mathbf{A}$ *is initialized to 0 and receives entrywise updates* $\mathbf{A}_{i,j} \leftarrow \mathbf{A}_{i,j} + \Delta$ *for* $\Delta \in \mathbb{R}$.

We initiate a systematic study of the dictionary learning and clustering problems in the setting where the assignment of the points to their sparse set of dictionary elements or clusters must be output together with the dictionary/cluster centers. Indeed, even for the popular Euclidean $k$-means clustering problem, almost all prior work that we are aware of only focus on outputting either only the cluster partitions, or the centers, but do not study the problem of recovering both. We address this problem by providing a dimensionality reduction technique that applies to $k$-means, sparse dictionary learning, and more generally to any problem of the form $\min_{\mathbf{X} \in \mathcal{X}, \mathbf{D} \in \mathbb{R}^{k \times d}} \|\mathbf{X}\mathbf{D} - \mathbf{A}\|_F^2$.

A typical approach for designing low-space streaming algorithms for clustering is to apply the standard Johnson–Lindenstrauss lemma [Johnson and Lindenstrauss, 1984, Boutsidis et al., 2010, Cohen et al., 2015, Becchetti et al., 2019, Makarychev et al., 2019]. This result states that if $\mathbf{G} \in \mathbb{R}^{d \times s}$ is an appropriately scaled dense sub-Gaussian matrix for $s = O(\epsilon^{-2} \log(k/\epsilon))$, then for any partition of $\mathbf{A}$ into $k$ clusters, the $k$-means clustering cost of $\mathbf{A}\mathbf{G}$ approximates the $k$-means clustering cost of $\mathbf{A}$ up to a $(1 \pm \epsilon)$ factor. Furthermore, $\mathbf{A}\mathbf{G}$ can be efficiently maintained in the turnstile streaming model (Definition 1.3) using just $ns = \tilde{O}(\epsilon^{-2}n)$ space, due to the linearity of the sketch $\mathbf{G}$. Note however that, naïvely, we cannot retrieve the corresponding centers of a clustering found by this method, since we have only stored the $s$-dimensional sketches of the $n$ points, and additional information must be stored in order to retrieve $d$-dimensional cluster centers which achieve a $(1 + \epsilon)$ approximation. In fact, we note in Theorem 4.1 that there is in fact a $\tilde{\Omega}(dk/\epsilon)$ space lower bound if we wish to output centers $\mathbf{C} \in \mathbb{R}^{k \times d}$ which achieve a $(1 + \epsilon)$ approximation, so the sketch $\mathbf{A}\mathbf{G}$ is *provably* insufficient for outputting both a nearly optimal assignment $\mathbf{X}$ and centers $\mathbf{C}$ when $n = \tilde{o}(\epsilon dk)$. We give a full discussion of our approaches for sketching and streaming algorithms for $k$ means clustering and dictionary learning and how we overcome this problem in Sections 2 and 3.

On the other hand, a study of lower bounds for the $k$-means clustering problem in the streaming setting when the assignment of points must be output is notably lacking in prior work as well. The main challenge in this setting is in obtaining the right dependence on $n$ and $\epsilon$. Indeed, an $\Omega(n)$ lower bound is immediate, since the size of the output is at least $\Omega(n)$ when we need to output assignments of the $n$ points to its appropriate cluster (in fact, we show in Theorems 4.3 and 4.4 that an $\Omega(n)$ lower bound follows even for outputting a constant factor approximation of the cost or centers). On the other hand, the previous upper bound using the Johnson–Lindenstrauss lemma to compute a nearly optimal assignment to clusters requires $\tilde{O}(\epsilon^{-2}n)$ bits of space. Note that there are many lower bounds that show that roughly $\epsilon^{-2}$ dimensions are required to apply the Johnson–Lindenstrauss lemma in various settings Nelson and Nguyên [2014], Kane et al. [2010], Larsen and Nelson [2016, 2017], Makarychev et al. [2019]. However, it is not clear whether or not this implies that $\epsilon^{-2}$ bits must be stored for all $n$ points in order to cluster them to a $(1 + \epsilon)$-approximately optimal clustering solution. Indeed, it may be possible that $\epsilon^{-2}$ bits are required only for much fewer than $n$ points, while the vast majority of the $n$ input points requires only $\tilde{O}(n)$ bits of space to assign to an approximately optimal center.

We present two lower bounds to partially address the question of impossibility results for assigning points to clusters in turnstile streams. Our main lower bound result is the following, which establishes an $\tilde{\Omega}(\epsilon^{-1}n)$ lower bound to output a $(1 + \epsilon)$-nearly optimal clustering. While this does not match the upper bound given by the Johnson–Lindenstrauss lemma, it shows that we cannot hope for a $\tilde{O}(n)$ upper bound in the turnstile streaming model in general.

**Theorem 1.1** (Informal restatement of Theorem C.1). *Let $k = d = \tilde{O}(1/\epsilon)$. Suppose a turnstile streaming algorithm outputs centers $\{\hat{c}^j\}_{j=1}^k \subseteq \mathbb{R}^d$ as well as assignments of $n$ points to the $k$ centers, which achieves a $(1 + \epsilon)$-approximately optimal solution to the $k$-means clustering problem. Then, the algorithm must use at least $\tilde{\Omega}(n/\epsilon)$ bits of space over any constant number of passes.*

As a second lower bound result, we also show that the Johnson-Lindenstrauss lemma is nearly tight if we require our algorithm to give a nearly optimal assignment of the input points to a fixed set of candidate centers. That is, we show in Theorem 4.2 that there is a fixed set of centers such that, if a turnstile streaming algorithm can assign each of the $n$ input points to a cluster such that the cost is at most $(1 + \epsilon)$ times the cost of the optimal assignment, then at least $\Omega(\epsilon^{-2}n)$ bits must be stored. A more detailed discussion of our lower bounds is given in Section 4.

Finally, we show that under some natural settings, one can obtain upper bounds that circumvent the lower bounds presented above. Indeed, we show that if we work in the *random order row arrival* streaming model, in which the input stream corresponds to the rows of $\mathbf{A}$ that arrive in a uniformly random order, then we can obtain upper bounds that depend on the *maximum sensitivity* of the input stream, and in particular, we obtain an upper bound using only $\tilde{O}(n)$ bits of space if the maximum sensitivity is sufficiently small (Theorem 4.5). Here, a bounded sensitivity assumption states that there are no points that can take up a significant fraction of the objective function, and can also be interpreted as a way to formalize a "well-clustered" instance.

## 2 Fixed parameter PTAS for sparse dictionary learning

### 2.1 PTAS for $r$-sparse dictionary learning

In this section, we provide an algorithm which solves the $r$-sparse dictionary learning problem (Definition 1.1) in time polynomial in the input matrix size ($n$) and dimension ($d$) up to $\epsilon$-relative error, for fixed $k$ and $\epsilon$. Additionally, we show that a similar approach can be used to provide an algorithm for $k$-means (Definition 1.2) that matches the current best dependency on $n, d, \epsilon$ and $k$ up to lower terms. First, we introduce a dimensionality reduction method that applies to both problems.

### 2.2 Dimensionality reduction

Our first step is to reduce the dimensionality of the given problem. Since the only difference between $k$-means and sparse dictionary learning is the constraint on the left factor, $\mathbf{X}$, we can use the same sketching approach to reduce both problems. Consider the following general definition:

**General problem**: Let $\mathcal{X} \subset \mathbb{R}^{n \times k}$ and $\mathbf{A} \in \mathbb{R}^{n \times d}$. Let $k \ll n, d$. Define the optimal solution as:

$$(\mathbf{X}^*, \mathbf{D}^*) = \underset{\mathbf{X} \in \mathcal{X}, \mathbf{D} \in \mathbb{R}^{k \times d}}{\operatorname{argmin}} \|\mathbf{X}\mathbf{D} - \mathbf{A}\|_F^2 \tag{1}$$

The following theorem states that one may efficiently reduce the dimensionality of $\mathbf{A}$ in sparse dictionary learning or $k$-means. We briefly sketch the ideas behind the reduction. Intuitively, the regression guarantee of Theorem 3.1 in Clarkson and Woodruff [2009] states that if $\mathbf{S}$ is a rank $k \ll d$ $\ell_2$-embedding matrix, then $\tilde{\mathbf{D}} = \operatorname{argmin}_{\mathbf{D} \in \mathbb{R}^{k \times d}} \|\mathbf{S}(\mathbf{X}^*\mathbf{D} - \mathbf{A})\|_F^2$ will be a good approximation to the optimal solution of the original problem. While we do not know $\mathbf{X}^*$, this guarantee implies that there is an approximately optimal dictionary, $\tilde{\mathbf{D}}$, in the row space of $\mathbf{S}\mathbf{A}$. We can then restrict the optimization problem to consider only dictionaries in this lower dimensional space. Therefore, we only need to consider the error residual in this lower dimensional space, so we may reduce the dimension of the problem by applying an affine-embedding matrix $\mathbf{T}$ and then applying SVD to find the dominant singular subspace of $\mathbf{S}\mathbf{A}\mathbf{T}$. Finally, we project the rows of $\mathbf{A}$ to this dominant subspace. We can then solve the lower dimensional problem and map the solution to the original space.

**Theorem 2.1.** *There is an algorithm which solves the problem in (1) up to $\epsilon \in (0,1)$ relative error with constant probability in $\mathcal{O}(\mathrm{nnz}(\mathbf{A}) + (n+d)\,\mathrm{poly}(k/\epsilon))$ time plus the time needed to solve:*

$$\min_{\mathbf{X} \in \mathcal{X}, \mathbf{D} \in \mathbb{R}^{k \times s}} \|\mathbf{X}\mathbf{D} - \mathbf{A}'\|_F^2,$$

*to within $\epsilon$-relative error for $s = \mathcal{O}(k \log(k)/\epsilon)$ and some $\mathbf{A}' \in \mathbb{R}^{n \times s}$ with constant probability.*

In the rest of this section, we assume that $d = \mathrm{poly}(k/\epsilon)$ for clearer exposition, since the above theorem implies we can reduce to this case efficiently.

## 2.3 Algorithm for sparse dictionary learning

The first component of our algorithm for sparse dictionary learning is a coreset construction that reduces the size of the problem from $n$ to a size that is logarithmic in $n$. We achieve this by first leveraging an existing coreset construction for projective clustering by Tukan et al. [2022]. In the $(\ell, m)$-projective clustering problem, the goal is to find a set of $\ell$ $m$-dimensional subspaces that minimizes the sum of the squared Euclidean distances of the input vectors $\{a^i\}_{i=1}^n$ to the closest subspace. Observe that, in the $r$-sparse dictionary problem, the minimum cost of a dictionary is the sum of the squared Euclidean distances of the input vectors to the $\binom{k}{r}$ subspaces spanned by any subset of $r$ vectors of the $k$ vectors in the dictionary. Hence, a coreset which preserves the projective clustering cost when $\ell = \binom{k}{r}$ will also preserve the cost of a dictionary in sparse dictionary learning.

After applying the coreset, we have reduced the size of the sparse dictionary problem to be at most logarithmic in $n$. This allows us to guess the sparsity pattern of the optimal left factor $\mathbf{X}^*$, since at most $r$ entries in each row of $\mathbf{X}^*$ may be nonzero. For each guess of the sparsity pattern of $\mathbf{X}^*$, we can find an approximately optimal solution under this constraint by recognizing this as a polynomial optimization problem. We apply the decision algorithm of Renegar [1992a] using binary search to determine each entry of $\mathbf{D}$ and the nonzero entries of $\mathbf{X}$ as done in Razenshteyn et al. [2016]. At some point we guess the sparsity pattern of $\mathbf{X}^*$, and hence attain an $\epsilon$-relative error solution to the sparse dictionary problem. The next theorem formally states the assumptions and guarantees of our algorithm, which is formalized in Algorithm 1 in the appendix.

**Theorem 2.2.** *For an input for the $r$-sparse dictionary learning problem (Definition 1.1) with error tolerance $\epsilon \in (0,1)$ such that the entries of $\mathbf{A}$ have bounded bit complexity, Algorithm 1 returns $\tilde{\mathbf{X}} \in \mathcal{X}$ and $\tilde{\mathbf{D}} \in \mathbb{R}^{k \times d}$ satisfying:*

$$\|\tilde{\mathbf{X}}\tilde{\mathbf{D}} - \mathbf{A}\|_F \leq (1 + \epsilon) \min_{\mathbf{X} \in \mathcal{X}, \mathbf{D} \in \mathbb{R}^{k \times d}} \|\mathbf{X}\mathbf{D} - \mathbf{A}\|_F,$$

*in $\mathrm{poly}(n)$ time with constant probability, when $k$, $r$, and $1/\epsilon$ are bounded by a constant.[1]*

## 2.4 Algorithms for $k$-means

The same general approach of applying dimensionality reduction and a coreset construction along with guessing the sparsity pattern of $\mathbf{X}^*$ can be used to achieve a fixed-parameter PTAS for $k$-means as well. However, we can achieve an improved time complexity matching the current best dependency on $k$ and $\epsilon$ up to lower order terms by further reducing the problem using results on leverage score sampling. Specifically, we combine Theorem 17 in Woodruff [2014b] and Theorem 3.1 in Clarkson and Woodruff [2009] to prove the following lemma.

**Lemma 2.1.** *There is a set of matrices $\mathcal{S} \subset \mathbb{R}^{s \times n}$ with exactly one non-zero entry per column such that for any $\mathbf{A} \in \mathbb{R}^{n \times k}$ and $\mathbf{B} \in \mathbb{R}^{n \times d}$, there exists $\mathbf{S} \in \mathcal{S}$, so that if:*

$$\tilde{\mathbf{X}} = \underset{\mathbf{X} \in \mathbb{R}^{k \times d}}{\mathrm{argmin}} \|\mathbf{S}(\mathbf{A}\mathbf{X} - \mathbf{B})\|_F \quad and \quad \mathbf{X}^* = \underset{\mathbf{X} \in \mathbb{R}^{k \times d}}{\mathrm{argmin}} \|\mathbf{A}\mathbf{X} - \mathbf{B}\|_F,$$

*then,*

$$\|\mathbf{A}\tilde{\mathbf{X}} - \mathbf{B}\|_F \leq (1 + \epsilon)\|\mathbf{A}\mathbf{X}^* - \mathbf{B}\|_F.$$

*Furthermore, $\mathcal{S}$ depends only on $n$, $k$, and $\epsilon$; and $|\mathcal{S}| = n^{\mathcal{O}(\frac{k \log k}{\epsilon})}$.*

---

[1] If $k$ and $r$ are not assumed to be constant, then the time complexity is $\exp((8k^{3r})^{O(k^{2r+1})} \log n)$.

After applying a coreset construction to reduce the $k$-means problem to size $\text{poly}(k/\epsilon)$, we can efficiently apply the above lemma to then reduce the problem to size $\tilde{\mathcal{O}}(k/\epsilon)$. Then, we brute force over all possible left-factors to find $\mathbf{X}^*$. The following theorem states our results formally.

**Theorem 2.3.** *For any input* $\mathbf{A} \in \mathbb{R}^{n \times d}$ *and* $\epsilon \in (0, 1)$*, Algorithm 2 will return a feasible solution to the $k$-means clustering problem (Definition 1.2),* $(\tilde{\mathbf{X}}, \tilde{\mathbf{D}})$*, satisfying:*

$$\|\tilde{\mathbf{X}}\tilde{\mathbf{D}} - \mathbf{A}\|_F \leq (1 + \epsilon) \cdot \min_{\mathbf{D} \in \mathbb{R}^{k \times d}, \mathbf{X} \in \mathcal{X}} \|\mathbf{X}\mathbf{D} - \mathbf{A}\|_F,$$

*with constant probability. Furthermore, Algorithm 2 runs in* $n \cdot \text{poly}(k/\epsilon) + \exp(\frac{k}{\epsilon} \text{polylog}(k/\epsilon))$ *time.*

## 3 Turnstile streaming algorithms

In this section, we consider the the *turnstile streaming model* (see Definition 1.3). We provide upper bounds on the space needed to compute an $\epsilon$-relative error solution to the $k$-means problem and a restricted form of the sparse dictionary learning problem in a turnstile stream. We do this by showing that these approximately optimal solutions can be computed from a few small linear sketches of the original data matrix, and any linear sketch can be trivially maintained in a turnstile stream by linearity of the updates. A key idea behind these algorithms is applying the *guess-the-sketch* approach introduced in Razenshteyn et al. [2016] along with the following theorem.

**Theorem 3.1.** *(Theorem 3.1 in Clarkson and Woodruff [2009]) Given* $\delta, \epsilon > 0$*, suppose* $\mathbf{A}$ *and* $\mathbf{B}$ *are matrices with* $n$ *rows, and* $\mathbf{A}$ *has rank at most* $k$*. There is an* $m = O(k \log(1/\delta)/\epsilon)$ *such that, if* $\mathbf{S}$ *is an* $m \times n$ *sign matrix, then with probability at least* $1 - \delta$*, if* $\tilde{\mathbf{X}} = \text{argmin}_{\mathbf{X}} \|\mathbf{S}(\mathbf{A}\mathbf{X} - \mathbf{B})\|_F^2$ *and* $\mathbf{X}^* = \text{argmin}_{\mathbf{X}} \|\mathbf{A}\mathbf{X} - \mathbf{B}\|_F^2$*, then* $\|\mathbf{A}\tilde{\mathbf{X}} - \mathbf{B}\|_F \leq (1 + \epsilon)\|\mathbf{A}\mathbf{X}^* - \mathbf{B}\|_F$*.*

Notice that, if we knew the optimal solution $\mathbf{X}^*$ exactly, then by the previous theorem we could compute an approximately optimal dictionary $\tilde{\mathbf{D}}$ exactly as $\tilde{\mathbf{D}} = (\mathbf{S}\mathbf{X}^*)^\dagger \mathbf{S}\mathbf{A}$. The key observation is that, since $\mathbf{S}$ is a random sign matrix and the rows of $\mathbf{X}$ are standard basis vectors, the set $\{\mathbf{S}\mathbf{X} \mid \mathbf{X} \in \mathcal{X}, \mathbf{S} \in \{\pm 1\}^{\tilde{\mathcal{O}}(k/\epsilon) \times n}\}$ is not too large. Also, we can approximately solve $\min_{\mathbf{X} \in \mathcal{X}} \|\mathbf{X}\tilde{\mathbf{D}} - \mathbf{A}\|_F^2$ for a fixed $\tilde{\mathbf{D}}$ with constant probability by solving $\tilde{\mathbf{X}} = \min_{\mathbf{X} \in \mathcal{X}} \|(\mathbf{X}\tilde{\mathbf{D}} - \mathbf{A})\mathbf{T}\|_F^2$, where $\mathbf{T}$ is a moderately sized affine embedding matrix. Since the number of possible $(\tilde{\mathbf{X}}, \tilde{\mathbf{D}})$ is not too large, an $\ell_2$-embedding matrix, $\mathbf{W}$, can be used to approximate $\|\tilde{\mathbf{X}}\tilde{\mathbf{D}} - \mathbf{A}\|_F^2$ for every possible $(\tilde{\mathbf{X}}, \tilde{\mathbf{D}})$.

Our streaming algorithm relies on carefully balancing the roles of the three sketching matrices to minimize the size of the sketches, using the weakest guarantee possible for each component. In particular, it is critical to use the affine embedding matrix $\mathbf{T}$ to only preserve the error for a fixed $\tilde{\mathbf{D}}$ instead of every subproblem and instead use the $\ell_2$-embedding matrix $\mathbf{W}$ to identify which subproblem provides an approximate solution to the overall problem.

**Theorem 3.2.** *(1) There are distributions of random sketching matrices* $\mathbf{T} \in \mathbb{R}^{d \times t}$*,* $\mathbf{S} \in \mathbb{R}^{s \times n}$*, and* $\mathbf{W} \in \mathbb{R}^{w \times nd}$*, with* $t = \mathcal{O}(\log(nk)/\epsilon^2)$*,* $s = \mathcal{O}(\frac{k}{\epsilon})$*, and* $w = \mathcal{O}(\frac{k^2}{\epsilon^3} \log(n))$ *such that* $\mathbf{S}\mathbf{A}$*,* $\mathbf{A}\mathbf{T}$*, and* $\mathbf{W}\text{vec}(\mathbf{A})$ *suffice to compute a* $(1 + \epsilon)$*-approximate solution to the $k$-means problem with at least constant probability, where* $\text{vec}(\mathbf{A}) \in \mathbb{R}^{nd}$ *is the flattening of* $\mathbf{A}$*.*

*(2) There is an algorithm which computes a* $(1+\epsilon)$*-approximate solution to the $k$-means problem in the turnstile model with at least constant probability using* $\tilde{\mathcal{O}}(n/\epsilon^2 + dk/\epsilon)$ *space for* $n, d > \text{poly}(k/\epsilon)$ *in* $n^{\tilde{\mathcal{O}}(k^2/\epsilon)}$ *additional time.*

The previous proof critically relies on the fact that $\{\mathbf{S}\mathbf{X} \mid \mathbf{X} \in \mathcal{X}, \mathbf{S} \in \{\pm 1\}^{m \times n}\}$ is a finite set that is not too large. We must therefore introduce the following restricted form of the sparse dictionary problem.

**Definition 3.1.** *(Discrete $r$-sparse dictionary problem) Let* $\mathcal{X}$ *be the space of* $n \times k$ *matrices with at most* $r$ *non-zero entries per row and non-zero entries taking values in* $\{-D, -(D - 1), ..., -1, 0, 1, ...(D - 1), D\}$*. The goal of this problem is solve the following optimization problem:*

$$\mathbf{X}^*, \mathbf{D}^* = \underset{\mathbf{X} \in \mathcal{X}, \mathbf{D} \in \mathbb{R}^{k \times d}}{\text{argmin}} \|\mathbf{X}\mathbf{D} - \mathbf{A}\|_F,$$

*where* $\mathbf{A} \in \mathbb{R}^{n \times d}$ *is an arbitrary input matrix.*

Under this constraint that the solution is in a discrete space the proof of the streaming algorithm for sparse dictionary learning proceeds essentially the same as for $k$-means while accounting for the larger solution space.

**Theorem 3.3.** *(1) There are distributions of random sketching matrices $\mathbf{T} \in \mathbb{R}^{d \times t}$, $\mathbf{S} \in \mathbb{R}^{s \times n}$, and $\mathbf{W} \in \mathbb{R}^{w \times nd}$, with $t = \mathcal{O}(r \log(nkD)/\epsilon^2)$, $s = \mathcal{O}(\frac{k}{\epsilon})$, and $w = \mathcal{O}(\frac{k^2}{\epsilon^3} \log(nD))$ such that $\mathbf{SA}$, $\mathbf{AT}$, and $\mathbf{W} \operatorname{vec}(\mathbf{A})$ suffice to compute a $(1 + \epsilon)$-approximate solution to the discrete $r$-sparse dictionary problem (Definition 3.1) with at least constant probability.*

*(2) There is an algorithm which computes a $(1 + \epsilon)$-approximate solution to the $r$-sparse dictionary problem in the turnstile model with at least constant probability using $\tilde{\mathcal{O}}(nr/\epsilon^2 + dk/\epsilon)$ space for $n, d > \operatorname{poly}(k/\epsilon)$ in $k^r \cdot (nD)^{\tilde{\mathcal{O}}(k^2/\epsilon)}$ additional time.*

Removing the restriction that $\mathbf{X}^*$ belongs to the restricted space would be an interesting future problem. However, two issues are that the entries of $\mathbf{X}$ may be very large, since the rows of $\mathbf{D}$ may not be orthogonal, and a uniform discretization is required to apply a guess-the-sketch argument.

# 4 Streaming lower bounds for Euclidean $k$-means clustering

We introduce slightly different definitions of the $k$-means clustering problem than the one used in Definition 1.2 to facilitate the notation of our lower bound arguments in this section.

**Definition 4.1** ($k$-means clustering cost). *Let $\{a^i\}_{i=1}^n \subseteq \mathbb{R}^d$ be a set of $n$ vectors in $d$ dimensions. Then, we define the $k$-means clustering cost of centers $c^1, c^2, \dots, c^k \in \mathbb{R}^d$ to be*

$$\operatorname{cost}(c^1, c^2, \dots, c^k) := \sum_{i=1}^n \min_{j=1}^k \|a^i - c^j\|_2^2.$$

**Definition 4.2** (Approximate solutions to $k$-means clustering). *Let $\{a^i\}_{i=1}^n \subseteq \mathbb{R}^d$ be a set of $n$ vectors in $d$ dimensions. Let*

$$\mathsf{OPT} := \min_{c^1, c^2, \dots, c^k \in \mathbb{R}^d} \operatorname{cost}(c^1, c^2, \dots, c^k)$$

*We say that an algorithm outputs an $\epsilon$-approximate solution to the $k$-means clustering problem if the algorithm outputs one of the following:*

- **Partition**: *a partition $C^1, C^2, \dots, C^k \subseteq [n]$ such that*

$$\sum_{j=1}^k \sum_{i \in C^j} \|a^i - \hat{c}^j\|_2^2 \le (1 + \epsilon)\mathsf{OPT}$$

  *where $\hat{c}^j := \frac{1}{|C^j|} \sum_{i \in C^j} a^i$.*

- **Centers**: *centers $\hat{c}^1, \hat{c}^2, \dots, \hat{c}^k \in \mathbb{R}^d$ such that $\operatorname{cost}(\hat{c}^1, \hat{c}^2, \dots, \hat{c}^k) \le (1 + \epsilon)\mathsf{OPT}$.*

- **Cost**: *a number $c \ge 0$ such that $\mathsf{OPT} \le c \le (1 + \epsilon)\mathsf{OPT}$.*

## 4.1 Lower bounds for $k$-means clustering

Our most technically involved and delicate lower bound result is the following theorem, which shows that nearly optimally solving $k$-means clustering to $(1 + \epsilon)$ accuracy requires $\tilde{\Omega}(n/\epsilon)$ bits of space:

**Theorem 1.1** (Informal restatement of Theorem C.1). *Let $k = d = \tilde{O}(1/\epsilon)$. Suppose a turnstile streaming algorithm outputs centers $\{\hat{c}^j\}_{j=1}^k \subseteq \mathbb{R}^d$ as well as assignments of $n$ points to the $k$ centers, which achieves a $(1 + \epsilon)$-approximately optimal solution to the $k$-means clustering problem. Then, the algorithm must use at least $\tilde{\Omega}(n/\epsilon)$ bits of space over any constant number of passes.*

We defer the full proof to Appendix C and give a proof sketch in this section to illustrate the most important ideas.

**The hard instance: set disjointness.** The starting point to our lower bound is the information theoretic communication complexity lower bound for the set disjointness problem due to Bar-Yossef et al. [2004]. In the two-party set disjointness problem, two players Alice and Bob each have a bit vector $A, B \in \{0, 1\}^d$ in $d$ dimensions, and they must determine whether there exists a coordinate $j \in [d]$ such that $A_j = B_j = 1$ or not. The work of Bar-Yossef et al. [2004] shows that in order to solve this problem, Alice and Bob must exchange messages that reveal at least $\Omega(d)$ bits of information about their inputs, which in turn implies an $\Omega(d)$ communication complexity lower bound for this problem, as well as an $\Omega(nd)$ communication complexity lower bound for solving a constant fraction of $n$ independent instances of the same problem. Furthermore, the hard instance of Bar-Yossef et al. [2004] has a simple input distribution: the vectors $(A, B)$ are such that the $j$th coordinate $(A^j, B^j)$ is drawn either as $(0, 0)$ with probability $1/2$ or $(1, 0)$ with probability $1/4$ or $(0, 1)$ with probability $1/4$, except for one coordinate, which may take the value $(1, 1)$.

We aim to make use of this result as follows. Consider the vector $Z = A + B$. This vector has entries in $\{0, 1\}$, except possibly for one entry, which could be 2. If we have $n$ such vectors, then we expect a good clustering into $k = d$ clusters to cluster all points with $Z_j = 2$ together. Such a clustering would be able to output the *index* of the intersection of $A$ and $B$, which intuitively requires more information than just determining whether there is an intersection or not, and thus should also require $\Omega(d)$ bits of information cost. Furthermore, we can choose the dimension $d$ to be roughly $1/\epsilon$, so that the cost of clustering $Z$ to the "correct" center will have a cost of $\Theta(d) = \Theta(1/\epsilon)$, while clustering $Z$ to the incorrect center will incur an additional error of $\Theta(1)$, which is an $\epsilon$ fraction of the cost.

**Cost calculations.** The main challenge in carrying out the idea in the previous paragraph is in arguing that the target optimal clustering that we wish to discover indeed is a nearly optimal clustering, and that significant deviations from this clustering result in a large cost. This involves showing a lower bound on the cost of *any* clustering.

Our first step is to obtain a lower bound on the cost of any clustering of $n$ random bit vectors in $d$ dimensions. If we first fix a set of $k$ centers $\{c^j\}_{j=1}^k$, then the minimum distance between a random bit vector $Z$ and any of the $c^j$ can be bounded by using Chernoff bounds, which implies a lower bound of $d/4 - O(\log d)$ on this quantity in expectation (Lemma C.4). Note, however, that this lower bound is not high enough to prevent a nearly optimal solution from just assigning points according to the best clustering of the random bits while ignoring the one entry that takes the value of $Z_j = 2$, which means that the clustering need not solve the problem of identifying the intersection coordinate between $A$ and $B$.

To address this problem, we need to make the cost of ignoring the intersection coordinate much more costly. We do this by instead considering the *multi-party* set disjointness problem, so that we now have $t = O(\sqrt{\log d})$ players rather than just 2, each with an input vector $A^{(i)} \in \{0, 1\}^d$, so that $Z = \sum_{i=1}^t A^{(i)}$ is now a random bit vector except for a single entry with a $t$ rather than a 2. Now, a clustering which does not correctly identify the intersection coordinate will pay a cost of roughly $t^2 = O(\log d)$, which is large enough to overcome the potential savings from a good clustering of the random bit coordinates. We also "plant" the target centers $c^j$ by adding roughly $n/k$ copies of each of our target centers $c^j$ as part of the input instance (Lemma C.7), so that choosing centers $\hat{c}^j$ that are significantly different from $c^j$ must incur a large cost. In particular, we can get the guarantee that on average, $\|c^j - \hat{c}^j\|_2^2 \le o(1)$.

At this point, we can argue that most of the $k$ centers are the centers that expect, i.e., roughly $t$ on one coordinate and $1/2$ on the rest of the coordinates. Thus, if we cluster a point $Z$ whose center we expect to be $\hat{c}^j$ but is clustered to some other $\hat{c}^{j'}$, and furthermore $\hat{c}^{j'}$ is close to our expected center $c^{j'}$, then we must incur an additional $O(\log d)$ cost which is too expensive. However, there is still the possibility that for the very small number of clusters $\hat{c}^j$ which do not satisfy $\|c^j - \hat{c}^j\|_2^2 \le o(1)$, these centers could be assigned a very large number of points with very low cost. We also show that this cannot be the case, by arguing that if a large number of points are assigned to very few clusters, then the cost must be large (Lemma C.8). With this lemma in hand, we are able to show our main result in Theorem C.1 by carefully combining the various cost contribution bounds discussed previously.

#### 4.1.1 Lower bound for outputting nearly optimal centers

We note that an $\Omega(dk/\epsilon)$ lower bound follows from an earlier lower bound for low rank approximation due to Woodruff [2014a], even for row arrival streams:

**Definition 4.3** (Row arrival stream). *We say that an algorithm outputs an $\epsilon$-approximate solution to the $k$-means clustering problem in the row arrival streaming model if the input vectors $\{a^i\}_{i=1}^n \subseteq \mathbb{R}^d$ arrive one at a time.*

**Theorem 4.1.** *Suppose that an algorithm outputs centers $\{\hat{c}^j\}_{j=1}^k \subseteq \mathbb{R}^d$ that achieves a $(1+\epsilon)$-approximately optimal solution to the $k$-means clustering problem after one pass through a row arrival stream (Definition 4.3). Then, the algorithm must use at least $\tilde{\Omega}(dk/\epsilon)$ bits of space.*

We briefly justify why the techniques of Woodruff [2014a] imply Theorem 4.1. The result of Woodruff [2014a] constructs a distribution over $O(k/\epsilon) \times d$ matrices such that one can recover an arbitrary random bit among $\tilde{\Omega}(dk/\epsilon)$ random bits by appending a set of $k$ "query" rows and then computing a $(1+\epsilon)$-approximately optimal low rank approximation to the resulting matrix. Furthermore, it is shown that a nearly optimal rank $k$ approximation is obtained by approximating all but $k$ rows by zero vectors. Such a rank $k$ approximation in fact corresponds to a clustering solution, and thus the proof of Woodruff [2014a] immediately applies to our $k$-means clustering setting as well.

### 4.2 Lower bounds for center cost query data structures

Next, we study lower bounds against streaming algorithms which have the guarantee of approximating the cost of an arbitrary but fixed set of centers. We formalize the guarantee we study in Definition 4.4.

**Definition 4.4** (Center cost query data structure). *We say that $\mathcal{Q}$ is an $\epsilon$-approximate center cost query data structure for the $k$ means clustering problem for the instance $\{a^i\}_{i=1}^n$ if, for any centers $c^1, c^2, \ldots, c^k \in \mathbb{R}^d$, $\mathcal{Q}$ outputs one of the following:*

- ***Partition***: *a partition $C^1, C^2, \ldots, C^k \subseteq [n]$ such that*

$$\sum_{j=1}^k \sum_{i \in C^j} \|a^i - c^j\|_2^2 \leq (1+\epsilon) \operatorname{cost}(c^1, c^2, \ldots, c^k).$$

- ***Cost***: *a number $c \geq 0$ such that*

$$\operatorname{cost}(c^1, c^2, \ldots, c^k) \leq c \leq (1+\epsilon) \operatorname{cost}(c^1, c^2, \ldots, c^k)$$

Our first lower bound is an $\Omega(n/\epsilon^2)$ bit space lower bound for a center cost query data structure which can output a partition for $k$-means clustering with $k = 2$. We proceed by a standard encoding argument, showing that any such data structure must encode $\Omega(n/\epsilon^2)$ many random bits. We provide the full proof in Appendix D.1.

**Theorem 4.2.** *Let $\epsilon \in (0, 1/3)$ and $k = 2$. Suppose that an algorithm maintains an $\epsilon/15$-approximate center cost query data structure for $k$-means clustering that outputs a partition (Definition 4.4) over a row arrival stream (Definition 4.3). Then, the algorithm must use at least $\Omega(n/\epsilon^2)$ bits of space, over any constant number of passes.*

### 4.3 Approximation of costs and centers

We show $\Omega(n)$ space memory bounds when we only need to estimate the optimal cost or centers achieving nearly optimal cost, up to a constant factor. Our lower bounds in this section are simpler reductions from the set disjointness problem Razborov [1990], Bar-Yossef et al. [2004]. Proofs are provided in Appendix D.2 and D.3.

**Theorem 4.3** (Lower Bound for Estimating $k$-Means Clustering Cost). *Let $k = 2$ and let $\mathcal{X}$ be the set of matrices $\mathbf{X} \in \mathbb{R}^{n \times k}$ with standard basis vectors as rows. Let $d = 1$. Any randomized algorithm which outputs a number $c \geq 0$ satisfying*

$$c \leq \min_{\mathbf{X} \in \mathcal{X}, \mathbf{D} \in \mathbb{R}^{k \times d}} \|\mathbf{X}\mathbf{D} - \mathbf{A}\|_F^2 < 2c \tag{2}$$

*in a constant number of passes over a turnstile stream requires $\Omega(n)$ bits of space.*

**Theorem 4.4** (Lower Bound for Computing Approximate Centers). *Let $k = 3$ and let $\mathcal{X}$ be the set of matrices $\mathbf{X} \in \mathbb{R}^{n \times k}$ with standard basis vectors as rows. Let $d = 1$. Any randomized algorithm which outputs centers $\tilde{\mathbf{D}} \in \mathbb{R}^{k \times d}$ satisfying*

$$\min_{\mathbf{X} \in \mathcal{X}} \|\mathbf{X}\tilde{\mathbf{D}} - \mathbf{A}\|_F^2 < 2 \min_{\mathbf{X} \in \mathcal{X}, \mathbf{D} \in \mathbb{R}^{k \times d}} \|\mathbf{X}\mathbf{D} - \mathbf{A}\|_F^2$$

*in a constant number passes over a turnstile stream requires $\Omega(n)$ bits of space.*

### 4.4 New upper bounds in random order streams

In this section, we show some new upper bounds showing that we can go beyond the previously presented lower bounds. In particular, in random order row arrival streams with bounded sensitivity, we show that the first segment of the stream is sufficient to obtain approximately optimal centers, and these can in turn be used to nearly optimally cluster the rest of the stream. We give the full proof of this result in Appendix D.4.

**Theorem 4.5.** *Suppose that the rows of $\mathbf{A} \in \mathbb{R}^{n \times d}$ arrive in a random order row arrival stream. Furthermore, suppose that the sensitivities of each row $a^i$ are bounded by $\alpha$, that is,*

$$\sup_{c^1, c^2, \ldots, c^k \in \mathbb{R}^d} \frac{\min_{j=1}^k \|a^i - c^j\|_2^2}{\sum_{i'=1}^n \min_{j=1}^k \|a^{i'} - c^j\|_2^2} \leq \alpha.$$

*Then, there is an algorithm which, with constant probability, outputs a $(1 + \epsilon)$-nearly optimal clustering with partitions and centers using*

$$\tilde{O}(\alpha nkd/\epsilon^4 + dk/\epsilon + n).$$

*bits of space. In particular, if $\alpha \leq \epsilon^4/kd$, then this algorithm uses just $\tilde{O}(n + dk/\epsilon)$ bits of space.*

## 5 Open directions

We conclude with several questions left open by our work.

1. In our PTAS for sparse dictionary learning of Theorem 2.2, can the bit complexity assumption be removed?

2. In the turnstile streaming setting, our main question is settling the space complexity of $k$-means clustering with assignments. Currently, the upper bound is $\tilde{O}(n/\epsilon^2)$ bits whereas our lower bound in Theorem C.1 is $\tilde{\Omega}(n/\epsilon)$ bits. Can this $\epsilon$ factor gap be closed by improving the upper bound or the lower bound?

3. In random order streaming model, we gave an $k$-means clustering upper bound using a bounded sensitivity assumption in Theorem 4.5. Can this assumption be removed? What upper bounds and lower bound are possible in this model?

## Acknowledgments and Disclosure of Funding

We thank the anonymous reviewers for useful feedback on improving the presentation of this work. Petros Drineas and Gregory Dexter were partially supported by NSF AF 1814041, NSF FRG 1760353, and DOE-SC0022085. David P. Woodruff and Taisuke Yasuda were supported by a Simons Investigator Award.

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
