# A Missing proofs for Section 2

In this section, we provide the missing proofs for Theorem 2.1, Theorem 2.2, and Theorem 2.3, along with prerequisite definitions and results. We also provide Algorithm 1 and Algorithm 2.

Recall that, after introducing the dimensionality reduction result of Theorem 2.1, we assume $d = \text{poly}(k/\epsilon)$ in subsequent sections for clearer exposition.

## A.1 Dimensionality reduction

We first restate an affine embedding guarantee provided for the CountSketch matrix by prior work.

**Lemma A.1.** *(From Lemma A.2 of Liu et al. [2020]) Given matrices* $\mathbf{A}$, $\mathbf{B}$ *with* $n$ *rows, a sparse embedding matrix* $\mathbf{S}$ *(i.e., CountSketch) with* $\mathcal{O}(\text{rank}(\mathbf{A})^2/\epsilon^2)$ *rows satisfies for all* $\mathbf{X}$ *of appropriate dimension with constant probability:*

$$\|\mathbf{S}(\mathbf{A}\mathbf{X} - \mathbf{B})\| = (1 \pm \epsilon)\|\mathbf{A}\mathbf{X} - \mathbf{B}\|_F^2$$

*Moreover, the matrix product* $\mathbf{S} \cdot \mathbf{A}$ *can be computed in* $\mathcal{O}(\text{nnz}(\mathbf{A}))$ *time.*

Next, we combine a few prior results to provide a regression error guarantee with a sketch that can be efficiently applied.

**Lemma A.2.** *Given* $\delta, \epsilon > 0$, *suppose* $\mathbf{A}$ *and* $\mathbf{B}$ *are matrices with* $n$ *rows, and* $\mathbf{A}$ *has rank at most* $k$. *There is an* $s = O(k \log(k)/\epsilon)$ *and a random matrix* $\mathbf{S} \in \mathbb{R}^{s \times n}$ *such that, with high constant probability, if:*

$$\tilde{\mathbf{X}} = \underset{\mathbf{X}}{\text{argmin}} \|\mathbf{S}(\mathbf{A}\mathbf{X} - \mathbf{B})\|_F^2 \quad \text{and} \quad \mathbf{X}^* = \underset{\mathbf{X}}{\text{argmin}} \|\mathbf{A}\mathbf{X} - \mathbf{B}\|_F^2,$$

*then,*

$$\|\mathbf{A}\tilde{\mathbf{X}} - \mathbf{B}\|_F \leq (1 + \epsilon)\|\mathbf{A}\mathbf{X}^* - \mathbf{B}\|_F.$$

*Furthermore,* $\mathbf{S} \cdot \mathbf{A}$ *can be computed in* $\mathcal{O}(\text{nnz}(\mathbf{A}) + d \cdot \text{poly}(k/\epsilon))$ *time.*

*Proof.* We will define $\mathbf{S} \in \mathbb{R}^{s \times n}$ as $\mathbf{S} = \mathbf{G} \cdot \mathbf{C}$, where $\mathbf{G} \in \mathbb{R}^{s \times c}$ is a Gaussian sketching matrix and $\mathbf{C} \in \mathbb{R}^{c \times n}$ is a CountSketch matrix, where $c = \text{poly}(k/\epsilon)$. Note that $\mathbf{SA}$ can be computed by first computing $\mathbf{CA}$ in $\mathcal{O}(\text{nnz}(\mathbf{A}))$ time and then computing $\mathbf{G} \cdot \mathbf{CA}$ in $\mathcal{O}(d \cdot \text{poly}(k/\epsilon))$ time.

Our first step is to show that the distribution of $\mathbf{S}$ is an $\ell_2$-subspace embedding (see Definition 2 of Woodruff [2014b]). By Theorem 9 of Woodruff [2014b], the distribution of $\mathbf{C}$ is an $\ell_2$-subspace embedding and by Theorem 6 of Woodruff [2014b], the distribution of $\mathbf{G}$ is an $\ell_2$-subspace embedding, each with high constant probability.

We can compose the $\ell_2$-subspace embedding guarantees to get the following bound with high probability via the union bound.

$$(1 - \epsilon)\|\mathbf{x}\|_2 \leq \|\mathbf{C}\mathbf{x}\|_2 \leq (1 + \epsilon)\|\mathbf{x}\|_2$$
$$\Rightarrow (1 - \epsilon)^2\|\mathbf{x}\|_2 \leq \|\mathbf{G}\mathbf{C}\mathbf{x}\|_2 \leq (1 + \epsilon)^2\|\mathbf{x}\|_2$$

Hence, $\mathbf{S}$ is an $\epsilon$-subspace embedding for a fixed $k$-dimensional space with high constant probability after adjusting $\epsilon$ by a constant factor. Therefore, $\|\mathbf{U}^T\mathbf{S}\mathbf{S}^T\mathbf{U} - \mathbf{I}\|_2 \leq \epsilon$, with high constant probability. The rest of the proof is the same as the proof of Theorem 3.1 in Clarkson and Woodruff [2009] while using this $\ell_2$-embedding matrix $\mathbf{S}$ instead of a random sign matrix. $\square$

## Proof of Theorem 2.1

*Proof.* By Lemma A.2, there exists a random matrix $\mathbf{S} \in \mathbb{R}^{s \times n}$ for $s = \mathcal{O}(\frac{k}{\epsilon} \log(k))$, such that, with at least constant probability,

$$\tilde{\mathbf{D}} = \underset{\mathbf{D} \in \mathbb{R}^{k \times d}}{\text{argmin}} \|\mathbf{S}(\mathbf{X}^*\mathbf{D} - \mathbf{A})\|_F^2 \Rightarrow \|\mathbf{X}^*\tilde{\mathbf{D}} - \mathbf{A}\|_F \leq (1 + \epsilon)\|\mathbf{X}^*\mathbf{D}^* - \mathbf{A}\|_F.$$

In this case, we can solve for $\tilde{\mathbf{D}}$ exactly as $\tilde{\mathbf{D}} = (\mathbf{SX}^*)^\dagger \mathbf{SA}$, hence, $\tilde{\mathbf{D}} = \mathbf{RSA}$ for some $\mathbf{R} \in \mathbb{R}^{k \times s}$. Therefore, $\tilde{\mathbf{D}} = \hat{\mathbf{R}}\mathbf{SA}$, where,

$$\hat{\mathbf{R}} = \underset{\mathbf{R} \in \mathbb{R}^{k \times s}}{\arg\min} \|\mathbf{X}^*\mathbf{RSA} - \mathbf{A}\|_F^2.$$

Let $\mathbf{T}_1 \in \mathbb{R}^{d \times \mathcal{O}(s^2/\epsilon^2)}$ be a count sketch matrix. Since $\mathrm{rank}(\mathbf{SA}) \leq s$, Lemma A.1 guarantees that $\|\mathbf{MSAT}_1 - \mathbf{AT}_1\|_F^2 = (1 \pm \epsilon)\|\mathbf{MSA} - \mathbf{A}\|_F^2$ for all $\mathbf{M} \in \mathbb{R}^{n \times s}$ simultaneously with at least constant probability. Since this holds for all $\mathbf{M} \in \mathbb{R}^{n \times s}$, and $\{\mathbf{XD} \mid \mathbf{X} \in \mathcal{X}, \mathbf{D} \in \mathbb{R}^{k \times s}\} \subset \mathbb{R}^{n \times s}$, we have that:

$$\tilde{\mathbf{X}}', \tilde{\mathbf{R}}' = \underset{\mathbf{X} \in \mathcal{X}, \mathbf{R} \in \mathbb{R}^{k \times s}}{\arg\min} \|\mathbf{XRSAT}_1 - \mathbf{AT}_1\|_F^2$$

$$\Rightarrow \|\tilde{\mathbf{X}}'\tilde{\mathbf{R}}'\mathbf{SA} - \mathbf{A}\|_F^2 \leq (1+\epsilon)\|\mathbf{X}^*\hat{\mathbf{R}}\mathbf{SA} - \mathbf{A}\|_F^2 = (1+\epsilon)\|\mathbf{X}^*\tilde{\mathbf{D}} - \mathbf{A}\|_F^2 \leq (1+\epsilon)^2\|\mathbf{X}^*\mathbf{D}^* - \mathbf{A}\|_F^2. \tag{3}$$

However, note that $\mathbf{SAT}_1$ has rank of at most $s$. Let $\mathbf{T}_2 \in \mathbb{R}^{\mathcal{O}(s^2/\epsilon^2) \times s}$ be the top $s$ right singular vectors of $\mathbf{SAT}_1$, and let $\mathbf{T} = \mathbf{T}_1\mathbf{T}_2$, then,

$$\begin{aligned}
\tilde{\mathbf{X}}', \tilde{\mathbf{R}}' &= \underset{\mathbf{X} \in \mathcal{X}, \mathbf{R} \in \mathbb{R}^{k \times s}}{\arg\min} \|\mathbf{XRSAT}_1 - \mathbf{AT}_1\|_F^2 \\
&= \underset{\mathbf{X} \in \mathcal{X}, \mathbf{R} \in \mathbb{R}^{k \times s}}{\arg\min} \|(\mathbf{XRSAT}_1 - \mathbf{AT}_1)\mathbf{T}_2\mathbf{T}_2^T\|_F^2 + \|\mathbf{AT}_1(\mathbf{I} - \mathbf{T}_2\mathbf{T}_2^T)\|_F^2 \\
&= \underset{\mathbf{X} \in \mathcal{X}, \mathbf{R} \in \mathbb{R}^{k \times s}}{\arg\min} \|\mathbf{XRSAT}_1\mathbf{T}_2 - \mathbf{AT}_1\mathbf{T}_2\|_F^2 \\
&= \underset{\mathbf{X} \in \mathcal{X}, \mathbf{R} \in \mathbb{R}^{k \times s}}{\arg\min} \|\mathbf{XRSAT} - \mathbf{AT}\|_F^2.
\end{aligned}$$

Notice that $\{\mathbf{RSAT} \mid \mathbf{R} \in \mathbb{R}^{k \times s}\} = \mathbb{R}^{k \times s}$ with probability one if $\mathrm{rank}(\mathbf{A}) > s$. If it does not hold that $\mathrm{rank}(\mathbf{A}) > s$, then we may directly reduce the dimension of the problem by SVD. Therefore, we can instead solve:

$$\tilde{\mathbf{X}}', \tilde{\mathbf{D}}' = \underset{\mathbf{X} \in \mathcal{X}, \mathbf{D} \in \mathbb{R}^{k \times s}}{\arg\min} \|\mathbf{XD} - \mathbf{AT}\|_F^2.$$

By the above equations, $\tilde{\mathbf{R}}' = \tilde{\mathbf{D}}'(\mathbf{SAT})^\dagger$ and by eqn. (3), $\|\tilde{\mathbf{X}}'\tilde{\mathbf{R}}'\mathbf{SA} - \mathbf{A}\|_F^2 \leq (1+\epsilon)^2\|\mathbf{X}^*\mathbf{D}^* - \mathbf{A}\|_F^2$. Therefore, we can return $\mathbf{X} = \tilde{\mathbf{X}}'$ and $\mathbf{D} = \mathbf{D}'(\mathbf{SAT})^\dagger\mathbf{SA}$ to guarantee:

$$\|\mathbf{XD} - \mathbf{A}\|_F^2 \leq (1+\epsilon)^2\|\mathbf{X}^*\mathbf{D}^* - \mathbf{A}\|_F^2 \leq (1+3\epsilon)\|\mathbf{X}^*\mathbf{D}^* - \mathbf{A}\|_F^2.$$

Now we work out the time complexity of the above reduction. First, we must compute $\mathbf{AT}$ to reduce to the smaller optimization problem. To do this, we can sample the CountSketch matrix $\mathbf{T}_1 \in \mathbb{R}^{k \times \mathcal{O}(s^2/\epsilon^4)}$ and compute $\mathbf{AT}_1$ in $\mathcal{O}(\mathrm{nnz}(\mathbf{A}) + \mathrm{poly}(k/\epsilon))$ time. Then, we sample the sketching matrix $\mathbf{S} \in \mathbb{R}^{\mathcal{O}(k/\epsilon \cdot \log k) \times n}$ and compute $\mathbf{SAT}_1$ in $\mathcal{O}(\mathrm{nnz}(\mathbf{A}) + \mathrm{poly}(k/\epsilon))$ time. Then, we compute $\mathbf{T}_2$ via the SVD of $\mathbf{SAT}_1$ and compute $\mathbf{AT} = \mathbf{AT}_1\mathbf{T}_2$ in $\mathrm{poly}(k/\epsilon)$ time. From here, we then solve the optimization problem for $\tilde{\mathbf{X}}'$ and $\tilde{\mathbf{D}}'$.

To convert $\tilde{\mathbf{D}}'$ to an approximate solution to the original problem, we must compute $\mathbf{D} = \mathbf{D}'(\mathbf{SAT})^\dagger\mathbf{SA}$. We can compute $(\mathbf{SAT})^\dagger$ via the SVD and then form $\mathbf{D}'(\mathbf{SAT})^\dagger$ in $\mathrm{poly}(k/\epsilon)$ time. Then, we compute the matrix product $\mathbf{SA}$ in $\mathcal{O}(\mathrm{nnz}(\mathbf{A}))$ time. Finally, the matrix product $\mathbf{D}'(\mathbf{SAT})^\dagger\mathbf{SA}$ can be computed in $\mathcal{O}(d \cdot \mathrm{poly}(k/\epsilon))$ time.

Therefore, the total time complexity of the reduction procedure is $\mathcal{O}(\mathrm{nnz}(\mathbf{A}) + (n + d)\,\mathrm{poly}(k/\epsilon))$. □

## A.2  PTAS for sparse-dictionary

### A.2.1  Coreset construction for Sparse Dictionary Learning

We begin by providing a coreset construction for the $r$-sparse dictionary learning problem, which we derive from coreset construction for the projective clustering problem defined here.

**Definition A.1.** *($(\ell, m)$-Projective clustering problem) Let $\mathbf{A} \in \mathbb{R}^{n \times d}$ be a matrix containing $n$ points. For a fixed sequence $\mathcal{F} = \{F_1, ..., F_\ell\}$, of $m$-dimensional subspaces, define:*

$$\text{cost}(\mathcal{F}, \mathbf{A}) = \sum_{i=1}^{n} \min_{F \in \mathcal{F}} \text{dist}(\mathcal{F}, \mathbf{A}_i)^2,$$

*where $\text{dist}(\mathbf{A}_i, F)^2$ denotes the squared Euclidean distance of the $i$-th row of $\mathbf{A}$ to the fixed subspace $F$.*

*The goal of the $(\ell, m)$-Projective clustering problem is to find a size $\ell$ collection of $m$-dimensional linear subspaces, $\mathcal{F}^*$, that minimizes the above cost function, i.e., $\mathcal{F}^* = \text{argmin}_{\mathcal{F}} \text{cost}(\mathcal{F}, \mathbf{A})$.*

We will use this to construct a reweighted form of the $r$-sparse dictionary problem with smaller size which we define next.

**Definition A.2.** *(Weighted $r$-SDL) Let $\mathcal{X}_r \subset \mathbb{R}^{n \times k}$ denote the set of matrices with at most $r$ non-zero entries per row. For a given input matrix $\mathbf{A} \in \mathbb{R}^{n \times d}$, such that $k \ll n, d$, and diagonal matrix $\mathbf{W} \in \mathbb{R}^{n \times n}$ return:*

$$(\mathbf{X}^*, \mathbf{D}^*) = \underset{\mathbf{X} \in \mathcal{X}, \mathbf{D} \in \mathbb{R}^{k \times d}}{\text{argmin}} \|\mathbf{W}(\mathbf{X}\mathbf{D} - \mathbf{A})\|_F^2. \tag{4}$$

*The parameter $k$ is the number of dictionary elements and the parameter $r$ determines how many dictionary elements can be used to represent each row of $\mathbf{A}$.*

**Theorem A.1.** *Let $r$, $k$, $\mathbf{A}$, and $\mathcal{X}$ be defined as in the sparse dictionary learning problem (Definition 1.1). If the entries of $\mathbf{A}$ can each be represented by $b$ bits, then there exists an algorithm which computes a diagonal matrix $\mathbf{W} \in \mathbb{R}^{w \times w}$ and $\mathbf{A}' \in \mathbb{R}^{w \times d}$ in $\mathcal{O}(n^2 k^{4r} b^{k^{2r+1}})$ time, such that,*

$$\left| \min_{\mathbf{X} \in \mathcal{X}} \|\mathbf{W}(\mathbf{X}\mathbf{D} - \mathbf{A}')\|_F^2 - \min_{\mathbf{X} \in \mathcal{X}} \|\mathbf{X}\mathbf{D} - \mathbf{A}\|_F^2 \right| \leq \epsilon \cdot \min_{\mathbf{X} \in \mathcal{X}} \|\mathbf{X}\mathbf{D} - \mathbf{A}\|_F^2,$$

*for all $\mathbf{D} \in \mathbb{R}^{k \times d}$. Furthermore, $w = \mathcal{O}((8k^{3r} b \log d)^{\mathcal{O}(k^{r+1})} \log n)$.*

*Proof.* First, we observe that any coreset for the $(\ell, m)$-projective clustering problem (Definition A.1) with $\ell = \binom{k}{r}$ and $m = r$ provides a coreset for the $r$-sparse dictionary learning problem. This is because if the collection of subspaces $\mathcal{F}$ contains all $r$-dimensional subspaces spanned by $r$ rows of the dictionary $\mathbf{D}$, then $\min_{\mathbf{X} \in \mathcal{X}} \|\mathbf{X}\mathbf{D} - \mathbf{A}\|_F^2 = \text{cost}(\mathcal{F}, \mathbf{A})$.

By Theorem 1.2[2] and Theorem 3.3 in Tukan et al. [2022], Algorithm 2 of Tukan et al. [2022] outputs a set of points $\mathcal{P}$ and weight function $w(p) : \mathcal{P} \to \mathbb{R}$ such that:

$$\left| \text{cost}(\mathcal{F}, \mathbf{A}) - \sum_{p \in \mathcal{P}} w(p) \cdot \min_{F \in \mathcal{F}} \text{dist}(\mathcal{F}, \mathbf{A}_i')^2 \right| \leq \epsilon \cdot \text{cost}(\mathcal{F}, \mathbf{A}),$$

for all $\mathcal{F}$ that are a $j$-size sequence of $k$-dimensional subspaces.

If $\mathcal{F}$ is the collection of all $r$-dimensional subspaces spanned by $r$ rows of the dictionary $\mathbf{D}$, then we can rewrite the above guarantee in matrix notation as follows:

$$\left| \min_{\mathbf{X} \in \mathcal{X}} \|\mathbf{W}(\mathbf{X}\mathbf{D} - \mathbf{A}')\|_F^2 - \min_{\mathbf{X} \in \mathcal{X}} \|\mathbf{X}\mathbf{D} - \mathbf{A}\|_F^2 \right| \leq \epsilon \cdot \min_{\mathbf{X} \in \mathcal{X}} \|\mathbf{X}\mathbf{D} - \mathbf{A}\|_F^2,$$

where $\mathbf{A}_i'$ is the $i$-th point in the point set $\mathcal{P}$ and $\mathbf{W} \in \mathbb{R}^{w \times w}$ is a diagonal matrix where $\mathbf{W}_{ii}$ is the weight $w(p_i)$.

Theorem 1.2 of Tukan et al. [2022] then guarantees that $w = \mathcal{O}((8\ell^3 \log(d\Delta))^{\mathcal{O}(\ell m)} \log n)$, where $\Delta$ is the the ratio of the largest and smallest non-zero entry magnitudes of $\mathbf{A}$. Therefore, $\Delta \leq 2^b$, and so $w = \mathcal{O}((8\ell^3 b \log d)^{\mathcal{O}(\ell m)} \log n)$. Furthermore, by the discussion below Theorem 3.3 of Tukan et al. [2022], their algorithm runs in $\mathcal{O}(n^2 \ell^4 (\log \Delta)^{\ell^2 m}) = \mathcal{O}(n^2 \ell^4 b^{\ell^2 m})$ time. Substituting in $\ell = k^r \geq \binom{k}{r}$ and $m = r$ to these bounds gives the final theorem statement.

$\square$

---

[2]We have confirmed through correspondence to the authors that there is a typo in Definition 1.9 of Tukan et al. [2022], and the definition should also state $(1 - \epsilon) \sum_{\mathbf{p} \in C} w(\mathbf{p}) \text{dist}(H(\mathbf{X}, \mathbf{v}), \mathbf{p})^2 \leq \sum_{\mathbf{p} \in C} \text{dist}(H(\mathbf{X}, \mathbf{v}), \mathbf{p})^2$. That is, Definition 1.9 defines a standard relative error coreset guarantee in the $\ell_2^2$-norm.

### A.2.2 Polynomial Solver for a Restricted SDL Problem

Next, we show that by adding a further restriction on the weighted $r$-SDL problem, we can solve the problem in polynomial time. First, define the *sparsity pattern* $\mathcal{N} \in \{(\mathcal{N}_i)_{i \in [n]} \mid |\mathcal{N}_i| = r, \mathcal{N}_i \subset [k]\}$, and let $\mathcal{X}_{\mathcal{N}}$ to be the set of $n \times k$ matrices such that $\mathbf{X}_{ij} = 0$ if $j \notin \mathcal{N}_{ij}$ for all $\mathbf{X} \in \mathcal{X}_{\mathcal{N}}$. That is, $\mathbf{X} \in \mathcal{X}_{\mathcal{N}}$ is a matrix where only $r$ fixed entries per row may be non-zero, and these entries are specified by the sparsity pattern $\mathcal{N}$. We define the following restricted solver.

**Definition A.3.** *For a given $r$-SDL problem, let* `PolySolver` *be an algorithm which takes as input a sparsity pattern $\mathcal{N}$, diagonal matrix $\mathbf{W} \in \mathbb{R}^{n \times n}$, input matrix $\mathbf{A} \in \mathbb{R}^{n \times d}$, dictionary size $k$, sparsity $r$, and error tolerance $\epsilon \in (0, 1)$. If $\mathcal{N}$ is the sparsity pattern of the optimal left-factor $\mathbf{X}^*$, then* `PolySolver` *outputs $\tilde{\mathbf{X}} \in \mathcal{X}_{\mathcal{N}}$ and $\tilde{\mathbf{D}} \in \mathbb{R}^{k \times d}$ which satisfy:*

$$\|\mathbf{W}(\tilde{\mathbf{X}}\tilde{\mathbf{D}} - \mathbf{A})\|_F^2 \le (1 + \epsilon) \cdot \|\mathbf{X}^*\mathbf{D}^* - \mathbf{A}\|_F^2.$$

**Lemma A.3.** *There exists an implementation of* `PolySolver` *that runs in $\mathcal{O}(2^{\mathcal{O}(nr+kd)})$ time given that the entries of $\mathbf{A}$ have bounded bit complexity.*

*Proof.* For $i \in [n]$ and $j \in [r]$, let $x_{ij}$ denote the $j$-th smallest entry in $\mathcal{N}_i$ of a matrix $\mathbf{X} \in \mathcal{X}_{\mathcal{N}}$. Observe that the entry $[\mathbf{X}\mathbf{D}]_{st}$ has the form $\sum_{j=1}^{r} x_{sj}\mathbf{D}_{\mathcal{N}_{s,j},t}$, hence $[\mathbf{W}(\mathbf{X}\mathbf{D} - \mathbf{A})]_{st} = \mathbf{W}_{ss}(\sum_{j=1}^{r} x_{sj}\mathbf{D}_{\mathcal{N}_{s,j},t} - \mathbf{A}_{st})$. Therefore, $\|\mathbf{W}(\mathbf{X}\mathbf{D} - \mathbf{A})\|_F^2$ is a fourth degree polynomial in the set of variables $\{x_{ij} \mid i \in [n], \ j \in [r]\}$ and the entries of $\mathbf{D}$.

By Renegar [1992a], for a given polynomial $P(y_1, y_2, ..., y_v)$ of degree $t$, we can determine whether there exists a solution satisfying $P(y_1, y_2, ..., y_v) \le L$ and $y_1^2 \le M$ in $(2t)^{\mathcal{O}(v)} \operatorname{poly}(H)$ time, where $H$ upper bounds the bit complexity of $L$ and $M$ (see Theorem 2.2 in Razenshteyn et al. [2016] for a restatement of this result). Under the assumptions of our lemma, $H$ is bounded by a constant.

We follow the approach of Razenshteyn et al. [2016] and use binary search to determine an approximately optimal solution for our polynomial minimization problem. First, since the bit complexity of the entries of $\mathbf{A}$ are assumed to be bounded by a constant, by Corollary 38 of Boutsidis et al. [2016], the objective error of the problem is either zero or greater than $2^{-\mathcal{O}(k)}$. Therefore, we can use binary search to find a value of $L$ satisfying $\|\mathbf{X}^*\mathbf{D}^* - \mathbf{A}\|_F^2 \le L \le (1 + \epsilon)\|\mathbf{X}^*\mathbf{D}^* - \mathbf{A}\|_F^2$ by running the decision algorithm of Renegar $\log 2^{\mathcal{O}(k)} = \mathcal{O}(k)$ times.

Then, we can repeatedly use binary search on each variable $y_i$ with the constraints $y_i^2 \le M$ and $P(y_1, y_2, ..., y_v) \le L$. After determining a variable $y_i$ through binary search, we can fix that variable, and then perform the procedure on the next variable. Overall, if the magnitude of the entries of $\mathbf{W}$, $\mathbf{X}^*$, and $\mathbf{D}^*$, are bounded by a doubly-exponential factor of $\mathcal{O}(nr + kd)$, we invoke the decision algorithm $2^{\mathcal{O}(nr+kd)}$ additional times to get an overall time complexity of $2^{\mathcal{O}(nr+kd)}$. $\square$

### A.2.3 Algorithm for sparse dictionary learning

Here, we present our algorithm for $r$-sparse dictionary learning along with a proof of its correctness and time complexity.

---

**Algorithm 1** PTAS for $r$-sparse dictionary learning

---

**Require:** $\mathbf{A} \in \mathbb{R}^{n \times d}$, $\epsilon \in (0, 1)$, and $k, r \in \mathbb{N}$ such that $r \le k$.
1: Compute $\mathbf{A}' \in \mathbb{R}^{w \times d}$ and $\mathbf{W} \in \mathbb{R}^{w \times w}$ by the algorithm of Theorem A.1.
2: Initialize $\tilde{\mathbf{D}} = \mathbf{0}$ and $\delta = \|\mathbf{A}\|_F$.
3: **for** $\mathcal{N} \in \{(\mathcal{N}_i)_{i \in [w]} \mid |\mathcal{N}_i| = r, \ \mathcal{N}_i \subset [k]\}$ **do**
4:     Compute $\mathbf{X}', \mathbf{D}' = $ `PolySolver`$(\mathcal{N}, \mathbf{W}, \mathbf{A}', k, r, \epsilon)$
5:     **if** $\|\mathbf{X}'\mathbf{D}' - \mathbf{W}\mathbf{A}'\|_F < \delta$ **then**
6:         Set $\tilde{\mathbf{D}} = \mathbf{D}'$ and $\delta = \|\mathbf{X}'\mathbf{D}' - \mathbf{W}\mathbf{A}'\|_F$
7:     **end if**
8: **end for**
9: **return** $\tilde{\mathbf{D}}$ and $\tilde{\mathbf{X}} = \operatorname{argmin}_{\mathbf{X} \in \mathcal{X}} \|\mathbf{X}\tilde{\mathbf{D}} - \mathbf{A}\|_F$.

---

**Proof of Theorem 2.2:**

*Proof. Correctness:* In Step 1 of the algorithm, by Theorem A.1, we compute the diagonal scaling matrix $\mathbf{W} \in \mathbb{R}^{w \times w}$ and $\mathbf{A}' \in \mathbb{R}^{w \times d}$ such that, for any fixed $\mathbf{D} \in \mathbb{R}^{k \times d}$:

$$\left| \min_{\mathbf{X} \in \mathcal{X}} \|\mathbf{W}(\mathbf{XD} - \mathbf{A}')\|_F^2 - \min_{\mathbf{X} \in \mathcal{X}} \|\mathbf{XD} - \mathbf{A}\|_F^2 \right| \leq \epsilon \cdot \min_{\mathbf{X} \in \mathcal{X}} \|\mathbf{XD} - \mathbf{A}\|_F^2.$$

Therefore, we can restrict our attention to solving for the dictionary $\mathbf{D}$ that minimizes the coreset error, $\min_{\mathbf{X} \in \mathcal{X}} \|\mathbf{W}(\mathbf{XD} - \mathbf{A}')\|_F^2$.

At some iteration of the loop, we will guess the sparsity pattern of $\mathbf{X}^* \in \mathcal{X}$, which we denote $\mathcal{N}^*$. By the guarantee of `PolySolver` (Definition A.3), $\mathbf{X}' \in \mathcal{X}_{\mathcal{N}^*}$ and $\mathbf{D}' \in \mathbb{R}^{k \times d}$ computed in Step 4 of the algorithm satisfy:

$$\|\mathbf{W}(\mathbf{X}'\mathbf{D}' - \mathbf{A}')\|_F^2 \leq (1 + \epsilon) \cdot \|\mathbf{X}^*\mathbf{D}^* - \mathbf{A}\|_F^2.$$

Therefore,

$$\min_{\mathbf{X} \in \mathcal{X}} \|\mathbf{XD}' - \mathbf{A}\|_F \leq (1 + \epsilon)^2 \cdot \min_{\mathbf{X} \in \mathcal{X}, \mathbf{D} \in \mathbb{R}^{k \times d}} \|\mathbf{XD} - \mathbf{A}\|_F.$$

Hence, the matrices $\tilde{\mathbf{D}}$ and $\tilde{\mathbf{X}}$ achieve $\epsilon$-relative error after adjusting by a constant factor.

*Time complexity:*

The overall time complexity of Algorithm 1 is given by:

$$\mathcal{O}(\text{Coreset construction}) + |\mathcal{N}| \times \texttt{PolySolver time} + \mathcal{O}(\text{Solve for } \mathbf{X})$$

By Theorem A.1, the coreset construction takes $\mathcal{O}(n^2 k^{4r} 2^{k^{2r+1}})$ time and $w = \mathcal{O}(((8k^{3r} b \log d)^{\mathcal{O}(k^{r+1})} \log n)$. The size of $\mathcal{N}$ is $|\mathcal{N}| = \binom{k}{r}^w$, and the time for one call to `PolySolver` is $\mathcal{O}(2^{\mathcal{O}(wr + \text{poly}(k/\epsilon))})$ by Lemma A.3. Therefore,

$$|\mathcal{N}| \times \texttt{PolySolver time} = \exp(w \cdot r \log k) \cdot \exp(wr) = \exp((8k^{3r} b \log d)^{\mathcal{O}(k^{r+1})} \log n)$$

Finally, solving for $\mathbf{X}$ takes $n \cdot \text{poly}(k, r, 1/\epsilon)$ time, so we can ignore this term. We conclude that, overall, Algorithm 1 runs in $\exp((8k^{3r} b \log d)^{O(k^{2r+1})} \log n)$ time. Note that this is equal to $\text{poly}(n)$ time under the assumption that $k, r, \epsilon$, and $b$ are bounded by a constant.

$\square$

## A.3 PTAS for $k$-means

In this section, we provide our algorithm for $k$-means along with a proof of its correctness and time complexity. In order to improve the time complexity dependency on $k$ and $\epsilon$, we use the idea of brute force leverage score sampling, which we introduce next.

### A.3.1 Brute force leverage score sampling

**Definition A.4.** *(Leverage Score Sampling - Definition 16 in Woodruff [2014b]) Let $\mathbf{Z} \in \mathbb{R}^{n \times k}$ have orthonormal columns, and let $p_i = \ell_i^2/k$, where $\ell_i^2 = \|\mathbf{e}_i^T \mathbf{Z}\|_2^2$ is the $i$-th leverage score of $\mathbf{Z}$. Note that $(p_1, ..., p_n)$ is a distribution. Let $\beta > 0$ be a parameter, and suppose we have any distribution $q = (q_1, ..., q_n)$ for which for all $i \in [n]$, $q_i \geq \beta p_i$.*

*Let $s$ be a parameter. Construct and $n \times s$ sampling matrix $\mathbf{\Omega}$ and an $s \times s$ rescaling matrix $\mathbf{D}$ as follows. Initially, $\mathbf{\Omega} = 0$ and $\mathbf{D} = 0$. For each column $j$ of $\mathbf{\Omega}, \mathbf{D}$, independently, and with replacement, pick a row index $i \in [n]$ with probability $q_i$, and set $\mathbf{\Omega}_{i,j} = 1$ and $\mathbf{D}_{jj} = 1/\sqrt{q_i s}$.*

**Lemma A.4.** *There is a set of matrices $\mathcal{S} \subset \mathbb{R}^{s \times n}$ with exactly one non-zero entry per column such that for any $\mathbf{A} \in \mathbb{R}^{n \times k}$ and $\mathbf{B} \in \mathbb{R}^{n \times d}$, there exists $\mathbf{S} \in \mathcal{S}$, so that if:*

$$\tilde{\mathbf{X}} = \underset{\mathbf{X} \in \mathbb{R}^{k \times d}}{\text{argmin}} \|\mathbf{S}(\mathbf{AX} - \mathbf{B})\|_F \quad and \quad \mathbf{X}^* = \underset{\mathbf{X} \in \mathbb{R}^{k \times d}}{\text{argmin}} \|\mathbf{AX} - \mathbf{B}\|_F,$$

*then,*

$$\|\mathbf{A}\tilde{\mathbf{X}} - \mathbf{B}\|_F \leq (1 + \epsilon)\|\mathbf{AX}^* - \mathbf{B}\|_F.$$

*Furthermore, $\mathcal{S}$ depends only on $n, k$, and $\epsilon$; and $|\mathcal{S}| = n^{\mathcal{O}(\frac{k \log k}{\epsilon})}$.*

*Proof.* Let $\mathbf{Z} \in \mathbb{R}^{n \times k}$ be a matrix with orthonormal columns. The corresponding leverage score sampling distribution $p$ satisfies $p_i = \|\mathbf{e}_i^T \mathbf{Z}\|_2^2 / k$. We can discretize each entry $p_i$ as follows. Let $\mathcal{I}_t = [1/2^{t-1}, 1/2^t)$. Then discretize each $p_i$ by setting $q_i = 1/2^{t-1}$ if $p_i \in \mathcal{I}_t$ for $t \leq \log n$, in which case $p_i \leq q_i \leq 2p_i$. If $p_i \notin \cup_{t \leq \log n} \mathcal{I}_t$, then set $q_i = \frac{2}{n}$, in which case $p_i < q_i$.

By Theorem 17 of Woodruff [2014b], if $\tilde{\mathbf{S}} = \mathbf{\Omega D}$ is constructed as described in Definition A.4 from the discretized distribution $q$, then for $s = \mathcal{O}(k \log(k)/\epsilon^2)$, with at least constant probability,

$$\|\mathbf{Z}^T \tilde{\mathbf{S}}^T \tilde{\mathbf{S}} \mathbf{Z} - \mathbf{I}\|_2 \leq \epsilon. \tag{5}$$

This implies that there exists a fixed matrix $\mathbf{S}$ with one non-zero entry per column achieving the above error guarantee that selects $s = \mathcal{O}(k \log(k)/\epsilon^2)$ rows of $\mathbf{Z}$ and rescales the row by $1/\sqrt{q_i s}$ when the $i$-th row is selected. Let $\mathcal{S}$ be the space of all matrices that select $s$ rows of $\mathbf{Z}$ with replacement and reweights the $i$-th row according to all possible configurations of $q$. Then, since there are $n^{\mathcal{O}(k \log k/\epsilon^2)}$ possible ways of selecting $s$ rows with replacement, and for a fixed selection of rows, the reweighting matrix $\mathbf{D}$ has $(\log n)^{\mathcal{O}(k \log(k)/\epsilon^2)}$ possibilities, $|S| = n^{\mathcal{O}(k \log(k)/\epsilon^2)}$.

At this point, we have shown that for parameter $\epsilon > 0$, there is a set of matrices $\mathcal{S}$ such that there exists $\mathbf{S} \in \mathcal{S}$ satisfying eqn. (5), and $|S| = n^{\mathcal{O}(k \log(k)/\epsilon^2)}$. By setting $\epsilon' = \sqrt{\epsilon}$ in the above result, and following the proof of Theorem 3.1 in Clarkson and Woodruff [2009], we can conclude the theorem statement. $\square$

### A.3.2 Algorithm for $k$-means

Here we present our fixed-parameter PTAS for $k$-means described in Section 2.4 and then provide the proof for Theorem 2.3.

---

**Algorithm 2** PTAS for k-means

---

**Require:** Input matrix $\mathbf{A} \in \mathbb{R}^{n \times d}$, error tolerance $\epsilon \in (0, 1)$, and number of clusters $k \in [n]$.
1: Compute a coreset for the $k$-means problem using Algorithm 3 of Bachem et al. [2018], denoted by the weights $\omega_1 ... \omega_n$, with $w = \text{poly}(k/\epsilon)$ non-zero weights.
2: Compute a $w \times n$ matrix $\mathbf{W}$, such that if $\omega_j$ is the $t$-th non-zero weight in the coreset, then $\mathbf{W}_{tj} = \omega_t$.
3: Initialize $\tilde{\mathbf{D}} = \mathbf{0}$ and $\delta = \|\mathbf{A}\|_F$.
4: **for** $\mathbf{S} \in \mathcal{S}_{w,k}$ **do**
5:     **for** $\mathbf{Y} \in \{\mathbf{SWX} \mid \mathbf{X} \in \mathcal{X}\}$ **do**
6:         Set $\mathbf{D}' = (\mathbf{SY})^\dagger \mathbf{SWA}$
7:         Compute $\mathbf{X}' = \text{argmin}_{\mathbf{X} \in \mathbf{W}\mathcal{X}} \|\mathbf{XD}' - \mathbf{WA}\|_F$ [3]
8:         **if** $\|\mathbf{X}'\mathbf{D}' - \mathbf{WA}\|_F < \delta$ **then**
9:             Set $\tilde{\mathbf{D}} = \mathbf{D}'$ and $\delta = \|\mathbf{X}'\mathbf{D}' - \mathbf{WA}\|_F$
10:         **end if**
11:     **end for**
12: **end for**
13: **return** $\tilde{\mathbf{D}}$ and $\tilde{\mathbf{X}} = \text{argmin}_{\mathbf{X} \in \mathcal{X}} \|\mathbf{X}\tilde{\mathbf{D}} - \mathbf{A}\|_F$.

---

**Proof of Theorem 2.3:**

*Proof. Correctness:*

In the first two steps of Algorithm 2, we use Algorithm 3 of Bachem et al. [2018] to compute an $\epsilon$-relative error coreset for $k$-means error. By Theorem 2 in Bachem et al. [2018], for some $w = \text{poly}(k/\epsilon)$, Algorithm 3 of Bachem et al. [2018] generates an epsilon relative error coreset with high constant probability. In matrix notation, this implies that their algorithm can be used to compute a matrix $\mathbf{W} \in \mathbb{R}^{w \times n}$ with one non-zero entry per row such that, for all $\mathbf{D} \in \mathbb{R}^{k \times d}$,

$$\left| \min_{\mathbf{X} \in \mathcal{X}} \|\mathbf{W}(\mathbf{XD} - \mathbf{A})\|_F - \min_{\mathbf{X} \in \mathcal{X}} \|\mathbf{XD} - \mathbf{A}\|_F \right| \leq \epsilon \cdot \min_{\mathbf{X} \in \mathcal{X}} \|\mathbf{XD} - \mathbf{A}\|_F.$$

---
[3]Let $\mathbf{W}\mathcal{X}$ denote the set $\{\mathbf{WX} \mid \mathbf{X} \in \mathcal{X}\}$, for the computed matrix $\mathbf{W}$.

Therefore, if $\mathbf{D}' \in \mathbb{R}^{k \times d}$ achieves less than $(1 + \epsilon)$ error on the coreset problem, then it will attain $(1 + \epsilon)^2 \leq 1 + 3\epsilon$ error on the original problem as well. By Lemma 2.1, when $\mathbf{Y} = \mathbf{SWX}^*$,

$$\mathbf{D}' = (\mathbf{SY})^\dagger \mathbf{SWA} = \underset{\mathbf{X} \in \mathbb{R}^{k \times d}}{\arg\min} \|\mathbf{S}(\mathbf{WX}^*\mathbf{D} - \mathbf{WA})\|_F,$$

which implies that,

$$\|\mathbf{WX}^*\mathbf{D}' - \mathbf{WA}\|_F \leq (1 + \epsilon) \cdot \min_{\mathbf{D} \in \mathbb{R}^{k \times d}} \|\mathbf{WX}^*\mathbf{D} - \mathbf{WA}\|_F.$$

Hence, in some iteration, $\mathbf{D}'$ will achieve at most a $1 + \epsilon$ factor error over the coreset problem, giving a $\epsilon$-relative error on the original problem after adjusting by a constant factor.

*Time complexity:*

First, by Lemma 2 of Bachem et al. [2018], computing $\mathbf{W}$ takes $\mathcal{O}(nkd)$ time.

Next, by Lemma 2.1, $|\mathcal{S}_{w,k}| = w^{\mathcal{O}(\frac{k \log k}{\epsilon})} = 2^{\mathcal{O}(\frac{k}{\epsilon} \operatorname{polylog}(k/\epsilon))}$. For a fixed $\mathbf{S} \in \mathcal{S}$, $|\{\mathbf{SWX} \mid \mathbf{X} \in \mathcal{X}\}| = k^{\mathcal{O}(\frac{k}{\epsilon} \log k)} = 2^{\mathcal{O}(\frac{k}{\epsilon} \operatorname{polylog}(k))}$, since there are $\mathcal{O}(\frac{k}{\epsilon})$ rows of $\mathbf{X}$ selected by $\mathbf{SW}$, and the non-zero entry in each of those rows can be in one of $k$ positions. This implies that the inner loop of Algorithm 2 is executed $\exp(\frac{k}{\epsilon} \operatorname{polylog}(k/\epsilon))$ times.

Hence, the overall running time is $n \cdot \operatorname{poly}(k/\epsilon) + \exp(\frac{k}{\epsilon} \operatorname{polylog}(k/\epsilon))$ under our assumption that $d = \operatorname{poly}(k/\epsilon)$. $\qquad\square$

# B  Information Theory Preliminaries

**Definition B.1** (Entropy and Mutual Information). *Let $X, Y, Z$ be discrete random variables. Then, the* entropy *of $X$ is defined as*

$$\mathsf{H}(X) := \sum_x \Pr[X = x] \log \frac{1}{\Pr[X = x]}$$

*and the* conditional entropy *of $X$ given $Y$ is defined as*

$$\mathsf{H}(X \mid Y) := \mathbb{E}_{y \sim Y}[\mathsf{H}(X \mid Y = y)]$$

*The* mutual information *between $X$ and $Y$ is defined as*

$$\mathsf{I}(X; Y) := \mathsf{H}(X) - \mathsf{H}(X \mid Y) = \mathsf{H}(Y) - \mathsf{H}(Y \mid X)$$

*and the* conditional mutual information *between $X$ and $Y$ given $Z$ is defined as*

$$\mathsf{I}(X; Y \mid Z) := \mathsf{H}(X \mid Z) - \mathsf{H}(X \mid Y, Z) = \mathsf{H}(Y \mid Z) - \mathsf{H}(Y \mid X, Z).$$

**Fact B.1** (Chain Rule). *Let $X_1, X_2, Y, Z$ be discrete random variables. Then,*

$$\mathsf{I}(X_1, X_2; Y \mid Z) = \mathsf{I}(X_1; Y \mid Z) + \mathsf{I}(X_2; Y \mid X_1, Z)$$

**Fact B.2.** *Let $X, Y$ be discrete random variables. Then, $\mathsf{H}(X) \geq \mathsf{H}(X \mid Y)$, with equality when $X$ and $Y$ are independent.*

**Lemma B.1** (Information cost decomposition (Lemma 5.1, Bar-Yossef et al. [2004])). *Let $\Pi$ be a protocol over $\mathcal{L}^n$ for some $\mathcal{L} \subseteq \mathcal{X} \times \mathcal{Y}$. Let $\zeta$ be a mixture of product distributions on $\mathcal{L} \times \mathcal{D}$, let $\eta = \zeta^n$, and suppose $((X, Y), D) \sim \eta$. Then, $\mathsf{I}(X, Y; \Pi(X, Y) \mid D) \geq \sum_{j=1}^{n} \mathsf{I}(X^j, Y^j; \Pi(X, Y) \mid D)$.*

## B.1  Total Variation Distance Lemma

We need the following total variation distance calculation:

**Lemma B.2** (Total variation distance bound). *Let $\mu$ be a distribution over a finite alphabet $Q$ and let $\mathcal{D} := \mu^d$. Let $\mathcal{D}'$ be the same distribution, except a uniformly random index $i \sim [d]$ is set to some $q^* \in Q$. Then,*

$$\mathsf{TV}(\mathcal{D}, \mathcal{D}') \leq \sqrt{\frac{1 - \mu(q^*)}{\mu(q^*)}} \frac{1}{\sqrt{d}}$$

*Proof.* For any $x \in Q^d$ and $q \in Q$, let

$$s_q(x) = |\{q \in Q : x_j = q\}|$$

denote the number of coordinates $j \in [d]$ such that $x_j = q$. Then, we have that

$$\mathcal{D}(x) = \prod_{q \in Q} \mu(q)^{s_q(x)}$$

$$\mathcal{D}'(x) = \sum_{x_j = q^*} \Pr(x \mid I = j) \Pr(I = j) = \frac{s_{q^*}(x)}{d} \frac{1}{\mu(q^*)} \prod_{q \in Q} \mu(q)^{s_q(x)}.$$

Then,

$$\begin{aligned}
\mathsf{TV}(\mathcal{D}, \mathcal{D}') &= \sum_{x \in Q^d} |\mathcal{D}(x) - \mathcal{D}'(x)| \\
&= \frac{1}{\mu(q^*)} \sum_{x \in Q^d} \mathcal{D}(x) \left| \mu(q^*) - \frac{s_{q^*}(x, y)}{d} \right| \\
&= \frac{1}{\mu(q^*)d} \sum_{x \in Q^d} \mathcal{D}(x) |s_{q^*}(x) - \mu(q^*)d| \\
&= \frac{1}{\mu(q^*)d} \mathop{\mathbb{E}}_{x \sim \mathcal{D}} [|s_{q^*}(x) - \mu(q^*)d|] \\
&\leq \frac{1}{\mu(q^*)d} \sqrt{\mathrm{Var}_{x \sim \mathcal{D}} [s_{q^*}(x)]} \\
&= \frac{1}{\mu(q^*)d} \sqrt{d \cdot \mu(q^*)(1 - \mu(q^*))} \\
&= \sqrt{\frac{1 - \mu(q^*)}{\mu(q^*)}} \frac{1}{\sqrt{d}}.
\end{aligned}$$

$\square$

# C  Proof of $\tilde{\Omega}(n/\epsilon)$ Lower Bound for $k$-Means Clustering

## C.1  Hardness Lemma for Assignment to Centers

In this section, we show information complexity lower bounds for a multi-player communication game based on a point assignment problem, when the input instance to the assignment problem is given by the sum $Z = \sum_{l=1}^{t} X^{(l)} \in \mathbb{R}^d$ of vectors $X^{(1)}, X^{(2)}, \dots, X^{(t)} \in \mathbb{R}^d$, each held by one of $t$ players, and we must assign $Z$ to the closest center $c^j \in \mathbb{R}^d$ for $j \in [k]$.

### C.1.1  Assignment of a Single Point

We start by studying the problem of assigning a single point to a set of centers, as well as a hard random instance for this problem. Our instance is based on the information theoretic approach to the set disjointness problem and its $t$-bit generalization due to Bar-Yossef et al. [2004]. We define the point assignment problem as follows:

**Definition C.1** (Point assignment problem). *Let $X^{(i)} \in \{0, 1\}^d$ be binary vectors for $i \in [t]$ such that $Z = \sum_{i=1}^{t} X^{(i)}$ has at most one entry $j \in [d]$ such that $Z_j > 1$. We say that a randomized protocol $\Pi(X^{(1)}, X^{(2)}, \dots, X^{(t)})$ solves the point assignment problem with probability at least $1 - \delta$ if for any $X^{(i)}$, $\Pi(X^{(1)}, X^{(2)}, \dots, X^{(t)})$ outputs some $e_j \in [d]$ such that $Z_j = t$ if such a $j \in [d]$ exists and any $e_l$ for $l \in [d]$ otherwise, with probability at least $1 - \delta$.*

The hard instance that we study for the point assignment problems is generated as follows. For each of the $d$ coordinates, with probability $1/2$, we set the $j$th coordinates of the $t$ players' vectors to all zeros, and with probability $1/2$, we set the $j$th coordinate of a uniformly random player to 1, and

everyone else's $j$th coordinate to 0. Finally, we select a uniformly random coordinate $j \in [d]$, and set the $j$th coordinate to 1 for every player with probability $1 - \alpha$ and 0 for every player with probability $\alpha$. The formal definition is given in Definition C.2:

**Definition C.2** (Hard instance for point assignment). *We define a distribution over $t$ random bit vectors in $d$ dimensions $\{X^{(i)}\}_{i=1}^t$ as follows. Let $B = \{B^j\}_{j=1}^d \sim [t]^d$, and let $I \sim [d]$ be a uniformly random index. Then for $j = I$, we draw the $j$th coordinates $\{X_j^{(i)}\}_{i=1}^t$ as*

$$C \sim \begin{cases} (1, 1, \ldots, 1) & w.p. \ 1 - \alpha \\ (0, 0, \ldots, 0) & w.p. \ \alpha \end{cases}$$

*and for $j \neq I$, we draw the $t$ values $\{X_j^{(i)}\}_{i=1}^t$ on the $j$th coordinate of each $X^{(i)}$ uniformly from $\{0, e_l\}$ where $l = B^j$. Let $\zeta$ denote the distribution over $(\{X^{(i)}\}_{i=1}^t, (I, B, C))$ on a single coordinate. We also denote by $Z$ the sum $Z = \sum_{i=1}^t X^{(i)} \in \mathbb{R}^d$.*

Throughout this section, we assume that $\Pi$ is a randomized protocol that solves the point assignment problem with probability at least $1 - \delta$. We now derive information complexity lower bounds for this problem, on the input instance of Definition C.2. We refer to Appendix B for standard preliminaries for information theory.

A crucial definition for the proof of the set disjointness information complexity lower bound of Bar-Yossef et al. [2004], as well as our point assignment lower bound, is the following:

**Definition C.3** (Conditional information complexity (Definition 4.5, Bar-Yossef et al. [2004])). *The $\delta$-error conditional information complexity of a function $f : \mathcal{X}^t \to \mathcal{Y}$ with respect to a distribution $\zeta$, denoted by $\mathsf{CIC}_{\zeta,\delta}(f)$, is defined as the smallest value of $\mathsf{I}(\{X^{(l)}\}_{l=1}^t; \Pi(\{X^{(l)}\}_{l=1}^t) \mid T)$ over the input distribution $(\{X^{(l)}\}_{l=1}^t, T) \sim \zeta$ for any $\delta$-error protocol $\Pi$ for $f$, that is, a protocol $\Pi$ which errs with probability at most $\delta$ on any input.*

We first show in Lemma C.1 that for $\Omega(d)$ coordinates $j \in [d]$, the $j$th coordinate must reveal $\Omega(1/t^2)$ bits of information, by lower bounding the information cost on the $j$th coordinate by the conditional information complexity of the $t$-bit AND problem, that is, $\mathrm{AND}_t(x^{(1)}, x^{(2)}, x^{(t)}) := \bigwedge_{l=1}^t x^{(l)}$. This conditional information complexity term is bounded by $\Omega(1/t^2)$ by Theorem 7.2 of Bar-Yossef et al. [2004]. As done in Bar-Yossef et al. [2004], the only valid inputs to the $\mathrm{AND}_t$ problem that we consider are the all 0 vector, the all 1 vector, and the $t$ standard basis vectors $e_l \in \{0, 1\}^t$ for $l \in [t]$.

**Lemma C.1** (Reduction lemma). *For at least $d/3$ coordinates $j \in [d]$,*

$$\mathsf{I}(\{X_j^{(l)}\}_{l=1}^t; \Pi(\{X^{(l)}\}_{l=1}^t) \mid I, B, C) \geq \frac{\alpha}{2} \mathsf{CIC}_{\zeta,\delta'}(\mathrm{AND}_t)$$

*for $\delta' := 4(3\delta + 2/\sqrt{d-1}) + (3/d + \sqrt{2t}/\sqrt{d})$, where $\zeta$ is the distribution defined in Definition C.2.*

*Proof.* Our proof roughly follows Lemma 5.2 of Bar-Yossef et al. [2004].

**Identifying $d/3$ good coordinates.** We first show that for a large number of coordinates $j \in [d]$, the protocol $\Pi$ is correct for the $\mathrm{AND}_t$ problem when restricted to the $j$th coordinate, that is, $\Pi$ outputs coordinate $j$ when $I = j$ and $C = (1, 1, \ldots, 1)$, while $\Pi$ outputs a coordinate other than $j$ when $C \neq (1, 1, \ldots, 1)$.

For $j \in [d]$, let $\delta(j)$ denote the failure probability of the protocol $\Pi$ over the input distribution of Definition C.2, conditioned on $I = j$. By averaging, we have that $\delta(j) \leq 3\delta$ for at least $(2/3)d$ coordinates $j \in [d]$. Next, for $j \in [d]$, let $p(j)$ denote the probability that the protocol $\Pi$ outputs the standard basis vector $e_j$, conditioned on $I = j$ and $C \neq (1, 1, \ldots, 1)$. First, if the input distribution is just the product distribution with each coordinate drawn as $\{X_j^{(i)}\}_{i=1}^t$ for $(\{X_j^{(i)}\}_{i=1}^t, D^j) \sim \zeta$, then note that at least $(2/3)d$ coordinates $j \in [d]$ will have $e_j$ output with probability at most $3/d$. Now if instead we uniformly draw $I \sim [d]$ and set $\{X_I^{(i)}\}_{i=1}^t = C$ for some $C \neq (1, 1, \ldots, 1)$, then the total variation distance between this distribution and the product distribution is at most $\sqrt{2t}/\sqrt{d}$ by a total variation distance calculation carried out in Lemma B.2. Thus, $p(j) \leq 3/d + \sqrt{2t}/\sqrt{d}$ for these $(2/3)d$ coordinates $j$. Now by a union bound, there are at least $d/3$ coordinates such that $\delta(j) \leq 3\delta$ and $p(j) \leq 3/d + \sqrt{2t}/\sqrt{d}$. We will show the information complexity lower bound on these coordinates. From this point forth in this proof, we fix $j$ to be such a coordinate.

**Reduction lemma.** Note that for any $j \in [d]$,

$$I(\{X_j^{(l)}\}_{l=1}^t; \Pi(\{X^{(l)}\}_{l=1}^t) \mid I, B, C)$$

$$= \mathop{\mathbb{E}}_{i \sim I, b^{-j} \sim B^{-j}} \left[ I(\{X_j^{(l)}\}_{l=1}^t; \Pi(\{X^{(l)}\}_{l=1}^t) \mid I = i, B^j, B^{-j} = b^{-j}, C) \right]$$

$$\geq \alpha \mathop{\mathbb{E}}_{i \sim I, b^{-j} \sim B^{-j}} \left[ I(\{X_j^{(l)}\}_{l=1}^t; \Pi(\{X^{(l)}\}_{l=1}^t) \mid I = i, B^j, B^{-j} = b^{-j}, C = (0, 0, \ldots, 0)) \right]$$

where the last inequality is true since $C = (0, 0, \ldots, 0)$ with probability $\alpha$.

Next, for each pair $(i, b^{-j})$, we construct a protocol $\Pi_{i,b}$ for a single copy of the AND problem with conditional information complexity loss exactly equal to

$$I(\{X_j^{(l)}\}_{l=1}^t; \Pi(\{X_j^{(l)}\}_{l=1}^t) \mid I = i, B^j, B^{-j} = b^{-j}, C = (0, 0, \ldots, 0)).$$

Let $\{x^{(l)}\}_{l=1}^t$ be a single copy of the $t$-bit AND problem. First note that conditioned on $I$, $B^{-j}$, and $C$, the hard instance of Definition C.2 is a product distribution, that is, the $t$ players can generate their inputs independently for all coordinates except $j$. Then, the $t$ players generate such an input instance according to $I = i$, $B^{-j} = b^{-j}$, and $C = (0, 0, \ldots, 0)$, and then replaces the $j$th input by $\{x^{(l)}\}_{l=1}^t$. The $t$ players then simulate the original protocol $\Pi$ with this input, and outputs 1 as the answer to the AND problem if $\Pi$ assigns the $j$th standard basis vector to $Z = \sum_{l=1}^t X^{(l)}$, and 0 otherwise.

Note that if $\{x^{(l)}\}_{l=1}^t$ is drawn according to the distribution of $C$ in Definition C.2 and the index $i$ on which to plant $C = (0, 0, \ldots, 0)$ is drawn uniformly randomly, then by Lemma B.2, the total variation distance between $\mathcal{D}$ conditioned on $I = j$ and the simulated input distribution $\mathcal{D}'$ is at most $2/\sqrt{d-1}$ (note that there are two different "$I$"s here, one for the original problem instance where we are setting the random coordinate $I = i$ to be all zeros, and one for the fixed coordinate $I = j$ to be the planted input $\{x^{(l)}\}_{l=1}^t$ in the simulated instance). Then, letting $S(\{X^{(l)}\}_{l=1}^t)$ be the event that the protocol $\Pi$ is successful on input $\{X^{(l)}\}_{l=1}^t$, we have that

$$\mathop{\Pr}_{\{X^{(l)}\}_{l=1}^t \sim \mathcal{D}'} [S(\{X^{(l)}\}_{l=1}^t)]$$

$$\geq \mathop{\Pr}_{\{X^{(l)}\}_{l=1}^t \sim \mathcal{D}} [S(\{X^{(l)}\}_{l=1}^t)] - \left| \mathop{\Pr}_{\{X^{(l)}\}_{l=1}^t \sim \mathcal{D}} [S(\{X^{(l)}\}_{l=1}^t)] - \mathop{\Pr}_{\{X^{(l)}\}_{l=1}^t \sim \mathcal{D}'} [S(\{X^{(l)}\}_{l=1}^t)] \right|$$

$$\geq \mathop{\Pr}_{\{X^{(l)}\}_{l=1}^t \sim \mathcal{D}} [S(\{X^{(l)}\}_{l=1}^t)] - \mathsf{TV}(\mathcal{D}, \mathcal{D}')$$

$$\geq 1 - 3\delta - \frac{2}{\sqrt{d-1}}.$$

Thus, $\Pi$ is successful with probability at least $1 - 3\delta - 2/\sqrt{d-1}$ under $\mathcal{D}'$. Then by averaging, we have that for at least $d/2$ choices of $I = i$, the $\Pi$ is successful with probability at least $1 - 2(3\delta + 2/\sqrt{d-1})$ conditioned on the choice of $I = i$.

Next, we bound the correctness probability of the protocol $\Pi_{i,b}$ for the $\mathrm{AND}_t$ problem, for the set of $d/2$ choices of $i$ as defined above. First, note that on this instance, if $\{x^{(l)}\}_{l=1}^t = (1, 1, \ldots, 1)$, then $\Pi$ is correct if and only if it assigns $Z$ to $e_j$, since $Z_j = t$ whereas $Z_l \leq 1$ for every other $l \in [d]$. Since $\Pi$ must be correct with probability at least $1 - 2(3\delta + 2/\sqrt{d-1})$ overall, it is correct with probability at least $1 - 4(3\delta + 2/\sqrt{d-1})$ conditioned on $\{x^{(l)}\}_{l=1}^t = (1, 1, \ldots, 1)$. On the other hand, if $\{x^{(l)}\}_{l=1}^t \neq (1, 1, \ldots, 1)$, then by our condition on the coordinate $j$, $\Pi$ assigns $e_j$ to $Z$ with probability at most $3/d + \sqrt{2t}/\sqrt{d}$. Thus, for these inputs, $\Pi_{i,b}$ is correct with probability at least $1 - (3/d + \sqrt{2t}/\sqrt{d})$. Thus, overall, $\Pi_{i,b}$ is correct with probability at least $1 - 4(3\delta + 2/\sqrt{d-1}) - (3/d + \sqrt{2t}/\sqrt{d}) = 1 - \delta'$ on any input.

Finally, let $(\{X'^{(l)}\}_{l=1}^t, B') \sim \zeta$. Then, note that the joint distribution of $(\{X'^{(l)}\}_{l=1}^t, B', \Pi_{i,b})$ is exactly the same as the joint distribution of $(\{X_j^{(l)}\}_{l=1}^t, B^j, \Pi(\{X^{(l)}\}_{l=1}^t))$, conditioned on $I = i, B^{-j} = b^{-j}, C = (0, 0, \ldots, 0)$. Thus, this shows that

$$I(\{X_j^{(l)}\}_{l=1}^t; \Pi(\{X^{(l)}\}_{l=1}^t) \mid I = i, B^j, B^{-j} = b^{-j}, C = (0, 0, \ldots, 0))$$

$$= I(\{X'^{(l)}\}_{l=1}^t; \Pi_{i,b}) \geq \mathsf{CIC}_{\zeta, \delta'}(\mathrm{AND}_t).$$

Chaining together the previous inequalities yields the claimed result. $\qquad\square$

Combining Lemma C.1 with Lemma B.1 yields the following:

**Lemma C.2.** *For $\delta \leq 1/50$ and $\sqrt{2t/d} \leq 1/20$, we have*

$$\mathsf{I}(\{X^{(l)}\}_{l=1}^t; \Pi(\{X^{(l)}\}_{l=1}^t) \mid I, B, C) = \Omega(d/t^2).$$

*Proof.* If $\delta \leq 1/50$, then $12\delta \leq 12/50 < 1/4$ so for large enough $d$, the $\delta'$ in Lemma C.1 is at most $1/3$. In this case, $\mathsf{CIC}_{\zeta,\delta'}(\mathrm{AND}_t) = \Omega(1/t^2)$ by Theorem 7.2 of Bar-Yossef et al. [2004], which, combined with Lemma B.1, yields the statement of the lemma. $\qquad\square$

### C.1.2 Assignment of Multiple Points

Next, we show by a direct sum argument that solving the assignment problem for $n$ points requires a protocol to reveal $\Omega(nd/t^2)$ bits of information.

**Lemma C.3.** *Let $\sqrt{2t/d} \leq 1/20$. Let $(\{X^{(l)}\}_{l=1}^t, (I, B, C)) = \{(\{X^{(i,l)}\}_{l=1}^t, (I^i, B^i, C^i))\}_{i=1}^n$ be drawn as $n$ i.i.d. from the hard distribution of Definition C.2. Suppose that a protocol $\Pi$ outputs a correct solution to the point assignment problem of Definition C.1 for least a $399/400$ fraction of points $\{X^{(i,l)}\}_{l=1}^t$ for $i \in [n]$, with probability at least $399/400$. Then,*

$$\mathsf{I}(\{X^{(l)}\}_{l=1}^t; \Pi(\{X^{(l)}\}_{l=1}^t) \mid I, B, C) = \Omega(nd/t^2).$$

*Proof.* Let $i \sim [n]$ be a uniformly random index. Then, by a union bound, the $i$th instance of the point assignment problem is solved correctly with probability at least $1 - 2/400 = 1 - 1/200$. Now for each fixed $i \in [n]$, let $\delta(i)$ be the probability that the $i$th instance is solved correctly. Then, over the randomness used by the protocol as well as $i \sim [n]$, we have that

$$\Pr_{i \sim [n]}\{i\text{th instance is correct}\} = \sum_{i=1}^n \frac{1}{n} \Pr\{i\text{th instance is correct}\} = \frac{1}{n}\sum_{i=1}^n 1 - \delta(i) \geq 1 - \frac{1}{200}$$

so $\mathbb{E}_{i \sim [n]}\delta(i) \leq 1/200$. Then for at least $n/2$ indices $i' \in [n]$, we have that $\delta(i') \leq 2/200 = 1/100$. We now claim that on these coordinates $i' \in [n]$, we have that

$$\mathsf{I}(\{X^{(i',l)}\}_{l=1}^t; \Pi(\{X^{(l)}\}_{l=1}^t) \mid I, B, C) = \Omega(d/t^2).$$

Indeed, note that $\mathsf{I}(\{X^{(i',l)}\}_{l=1}^t; \Pi(\{X^{(l)}\}_{l=1}^t) \mid I, B, C)$ is the expectation of

$$\mathsf{I}(\{X^{(i',l)}\}_{l=1}^t; \Pi(\{X^{(l)}\}_{l=1}^t) \mid I^{i'}, B^{i'}, C^{i'}, I^{-i'} = i^{-i'}, B^{-i'} = b^{-i'}, C^{-i'} = c^{-i'})$$

over $i^{-i'} \sim I^{-i'}, b^{-i'} \sim B^{-i'}, c^{-i'} \sim C^{-i'}$. Now for each fixing $i^{-i'}, b^{-i'}, c^{-i'}$, let $\delta(i^{-i'}, b^{-i'}, c^{-i'})$ that the $i'$th instance of the point assignment problem is correct given these fixings. Then by Markov's inequality, for at least half of the fixings, we have $\delta(i^{-i'}, b^{-i'}, c^{-i'}) \leq 2/100 = 1/50$. Note that each of these fixings corresponds to a protocol for solving the point assignment problem with probability at least $1 - 1/50$. Thus, we have by Lemma C.2 that

$$\mathsf{I}(\{X^{(i',l)}\}_{l=1}^t; \Pi(\{X^{(l)}\}_{l=1}^t) \mid I^{i'}, B^{i'}, C^{i'}, I^{-i'} = i^{-i'}, B^{-i'} = b^{-i'}, C^{-i'} = c^{-i'}) = \Omega(d/t^2)$$

for these fixings. Since this event occurs with probability at least $1/2$, it follows that $\mathsf{I}(\{X^{(i',l)}\}_{l=1}^t; \Pi(\{X^{(l)}\}_{l=1}^t) \mid I, B, C) = \Omega(d/t^2)$ as well.

Finally, by Lemma B.1, we have that

$$\mathsf{I}(\{X^{(l)}\}_{l=1}^t; \Pi(\{X^{(l)}\}_{l=1}^t) \mid I, B, C) \geq \sum_{i=1}^n \mathsf{I}(\{X^{(i,l)}\}_{l=1}^t; \Pi(\{X^{(l)}\}_{l=1}^t) \mid I, B, C)$$

$$\geq \frac{n}{2} \cdot \Omega(d/t^2) = \Omega(nd/t^2).$$

$\qquad\square$

## C.2 Lower Bounds for Clustering in Row Insertion Streams

Our first result is to show that an algorithm for computing a $(1 + \epsilon)$-approximate nearly optimal $k$-means clustering on $n$ points for $k = d = \Theta(1/\epsilon)$ on row insertion streams requires $\Omega(n/\epsilon)$ bits of space.

For this result, we need a lower bound against any nearly optimal clustering, so we need to "plant" our desired centers in order to force the solution to look like standard basis vectors. This will allow us to use the clustering algorithm to solve the point assignment problem. In order to determine the number of points we need to plant the centers, we first need a lower bound on the cost of any clustering of random bits, which we show in the next section.

### C.2.1 Cost Lower Bound on Random Points

We first lower bound the cost of any clustering of the random points of the hard instance in Definition C.2. We start with a bound in expectation:

**Lemma C.4** (Expectation bound for clustering random bits). *Fix a set of centers $c^1, c^2, \ldots, c^k \in [0,1]^d$. Let $Z \in \{0,1\}^d$ be a vector of $d$ uniformly random bits. Then,*

$$\mathop{\mathbb{E}}_{Z}\left[\min_{j=1}^{k} \|Z - c^j\|_2^2\right] \geq \frac{d}{4} - \frac{\log(kd) + 1}{2}.$$

*Proof.* Let $\mu := \mathbb{E}[Z]$ (i.e., the vector with $1/2$ in every entry). Fix a specific center $c^j$ for $j \in [k]$. Then,

$$\|Z - c^j\|_2^2 = \|Z - \mu\|_2^2 + \|\mu - c^j\|_2^2 + 2\langle Z - \mu, \mu - c^j\rangle = \frac{d}{4} + \|\mu - c^j\|_2^2 + 2\langle Z - \mu, \mu - c^j\rangle$$

By Hoeffding's inequality, we have

$$\Pr\{|\langle Z - \mu, \mu - c^j\rangle| \geq t\|\mu - c^j\|_2\} \leq 2\exp(-2t^2)$$

so for $t = \sqrt{\log(kd)/2}$, this probability is at most $2/kd$. By a union bound over the $k$ choices of $j$, we have that

$$\Pr\left\{\min_{j=1}^{k} \|Z - c^j\|_2^2 \leq \frac{d}{4} + \|\mu - c^j\|_2^2 - 2\sqrt{\log(kd)/2}\|\mu - c^j\|_2\right\} \leq \frac{2}{d}.$$

Note that

$$\|\mu - c^j\|_2^2 - 2\sqrt{\log(kd)/2}\|\mu - c^j\|_2 = \left(\|\mu - c^j\|_2 - \sqrt{\log(kd)/2}\right)^2 - \log(kd)/2 \geq -\log(kd)/2$$

so

$$\Pr\left\{\min_{j=1}^{k} \|Z - c^j\|_2^2 \geq \frac{d}{4} - \frac{\log(kd)}{2}\right\} \geq 1 - \frac{2}{d}.$$

It follows that

$$\mathbb{E}\left[\min_{j=1}^{k} \|Z - c^j\|_2^2\right] \geq \left(1 - \frac{2}{d}\right)\left(\frac{d}{4} - \frac{\log(kd)}{2}\right) \geq \frac{d}{4} - \frac{\log(kd) + 1}{2}$$

$\square$

Lemma C.4 shows that when clustering random bits, we can only save approximately a $(1 - 1/\tilde{\Theta}(d))$ factor for any clustering compared to a single center, in expectation. Since all but one coordinate in the hard instance of Definition C.2 are random bits, and the one coordinate can only decrease the cost by a factor of $(1 - 1/\tilde{\Theta}(d))$, any clustering into $k$ centers still has cost at least approximately $(1 - 1/\tilde{\Theta}(d))$ times the cost of a single center.

The next lemma converts the result of Lemma C.4 into a high probability result about any clustering, via a net argument.

**Lemma C.5.** *Let $\{Z^i\}_{i=1}^n$ be $n$ independent uniformly random bit vectors in $d$ dimensions. Suppose that $n \geq 16d\log(d^{2d}/\delta) = 32d^2\log(d/\delta)$. Then, with probability at least $1 - \delta$, we have that*

$$\min_{c^1,c^2,\ldots,c^k \in [0,1]^d} \sum_{i=1}^{n} \min_{j=1}^{k} \|Z^i - c^j\|_2^2 \geq n\left(\frac{d}{4} - \frac{\log(kd) + 9}{2}\right).$$

*Proof.* Let $\{c^j\}_{j=1}^k$ and $\{c'^j\}_{j=1}^k$ be two sets of centers such that $\|c^j - c'^j\|_2^2 \le 1/d$. Then,

$$\min_{j=1}^k \|Z^i - c^j\|_2^2 \le \min_{j=1}^k \|Z^i - c'^j\|_2^2 + \|c'^j - c^j\|_2^2 + 2\|Z^i - c'^j\|_2 \|c'^j - c^j\|_2$$

$$\le \min_{j=1}^k \|Z^i - c'^j\|_2^2 + 3$$

so if $\{c^j\}_{j=1}^k$ has high cost, then $\{c'^j\}_{j=1}^k$ must as well. We now consider a net $\mathcal{N} \subseteq [0,1]^d$ of size $d^{2d}$ such that for any $c \in [0,1]^d$, there exists $c' \in \mathcal{N}$ such that $\|c - c'\|_2^2 \le 1/d$. Now fix a set of centers $\{c^j\}_{j=1}^k \in \mathcal{N}^k$. By Lemma C.4, we have that

$$\mathbb{E}_Z[\min_{j=1}^k \|Z - c^j\|_2^2] \ge \frac{d}{8}$$

for sufficiently large $d$, so we have that

$$\Pr\left\{\sum_{i=1}^n \min_{j=1}^k \|Z^i - c^j\|_2^2 \le (1 - 1/d)n \, \mathbb{E}_Z\left[\min_{j=1}^k \|Z - c^j\|_2^2\right]\right\} \le \exp\left(-\frac{nd}{16d^2}\right) \le \frac{\delta}{d^{2d}}$$

by Chernoff bounds. Then by a union bound, the same holds simultaneously for every $\{c^j\}_{j=1}^k \in \mathcal{N}^k$ with probability at least $1 - \delta$.

Now for an arbitrary set of centers $c^1, c^2, \ldots, c^k \in [0,1]^d$, there exists some $\{c'^j\}_{j=1}^k \in \mathcal{N}^k$ such that $\|c^j - c'^j\|_2^2 \le 1/d$ for every $j \in [k]$. Then,

$$\sum_{i=1}^n \min_{j=1}^k \|Z^i - c^j\|_2^2 \ge \sum_{i=1}^n \left(\min_{j=1}^k \|Z^i - c'^j\|_2^2 - 3\right)$$

$$\ge (1 - 1/d)n \, \mathbb{E}_Z\left[\min_{j=1}^k \|Z - c'^j\|_2^2\right] - 3n$$

$$\ge (1 - 1/d)n\left(\frac{d}{4} - \frac{\log(kd) + 1}{2}\right) - 3n$$

$$\ge n\left(\frac{d}{4} - \frac{\log(kd) + 9}{2}\right).$$

$\square$

### C.2.2 Upper Bound on a Nearly Optimal Cost

We first upper bound the optimal cost of clustering by giving an explicit clustering construction, and upper bounding the cost. We define this clustering in Definition C.4:

**Definition C.4** (Nearly optimal clustering). *We define a clustering for points drawn from Definition C.2. Consider the variables $I$ and $C$ as defined in Definition C.2. If $C = (1, 1, \ldots, 1)$ and $I = j$, then we assign the point to cluster $j$. On the other hand, if $C \ne (1, 1, \ldots, 1)$ and $I = j$, then we assign the point to a uniformly random point $j' \in [d] \setminus \{j\}$ such that $X_{j'}^{(l)} = 1$ for some $l \in [t]$. If no such coordinate exists, we assign it to any cluster. Furthermore, we define the center $c^j$ by setting its $j'$th coordinate to be*

$$c_{j'}^j = \begin{cases} \frac{t+1}{2} & \text{if } j' = j \\ \frac{1}{2} & \text{if } j' \ne j \end{cases}$$

The cost of this clustering is bounded in the following lemma:

**Lemma C.6.** *Let $\{Z^i\}_{i=1}^n$ be drawn i.i.d. from the distribution of Definition C.2. Then, with probability at least $1 - (1/2)^{d-1}$, the clustering defined in Definition C.4 has cost at most $n(d + t^2 - 2t)/4$.*

*Proof.* Let $(\{X^{(i,l)}\}_{l=1}^t, (I^i, B^i, C^i))$ denote the $i$th element drawn from Definition C.2, for $i \in [n]$. We handle the cost calculation by conditioning on the event that at least one nonzero coordinate is drawn on $[d] \setminus \{I^i\}$, since this occurs with probability at least $1 - (1/2)^{d-1}$.

Fix a cluster $j \in [k]$. We will consider the distribution of points $\{X^{(i,l)}\}_{l=1}^t$, conditioned on the event that the point being clustered to cluster $j$ in the clustering of Definition C.4. Note then that the $j$th coordinate comes from a point such that $I^i = j$ and $C^i = (1, 1, \dots, 1)$, or the $j$th coordinate comes from a point with $\sum_{l=1}^t X^{(i,l)} = 1$ and $I^i \neq j$ and $C^i = (0, 0, \dots, 0)$. In either case, the coordinates $[d] \setminus \{j\}$ are in $\{0, 1\}$, and the $j$th coordinate is in $\{1, t\}$. Then for our defined center $c^j$, the squared cost is $(1/2)^2 = 1/4$ on $d - 1$ coordinates and $((t-1)/2)^2 = (t-1)^2/4$ on one coordinate per point, for a total of $n \cdot ((d-1)/4 + (t-1)^2/4) = n(d + t^2 - 2t)/4$ as claimed. $\quad\square$

### C.2.3  Planting Centers

With our nearly optimal clustering of Definition C.4 in mind, we now add copies of these centers into our instance in order to encourage the clustering algorithm to find this solution. Note that this increases the cost of any other clustering, without increasing the cost of this clustering.

**Lemma C.7.** *Let $n \geq 32d^2 \log(d/\delta)$. Consider the input instance to $k$-means clustering given by $n$ random points drawn according to Definition C.2, together with*

$$\gamma := \frac{400t^2 n}{k} \left( \frac{\log(kd) + 9}{2} + \frac{t^2 - 2t}{4} + \frac{(d + t^2 - 2t)}{4d} \right) = O\left( \frac{t^2 n}{k} (\log(kd) + t^2) \right)$$

*copies of each center $c^j$ for $j \in [k]$ as defined in Definition C.4. Furthermore, let $\{\hat{c}^j\}_{j=1}^k$ be centers achieving a $(1 + 1/d)$-nearly optimal solution to the $k$-means clustering instance. Then, $\|c^j - \hat{c}^j\|_2^2 \leq 1/4$ for at least $(1 - 1/100t^2)k$ of the centers $c^j$.*

*Proof.* Recall that in Lemma C.5, we showed that any clustering of $n$ random points drawn from Definition C.2 must have a cost of at least $nd/4 - n(\log(kd) + 9)/2$ with probability at least $1 - \delta$. Then, with probability at least $1 - (1/2)^{d-1}$, the value of the optimal solution is bounded above by $n(d + t^2 - 2t)/4$ by Lemma C.6, so we must have that

$$\gamma \sum_{j=1}^k \|c^j - \hat{c}^j\|_2^2 + n\left( \frac{d}{4} - \frac{\log(kd) + 9}{2} \right) \leq (1 + 1/d)\frac{n(d + t^2 - 2t)}{4}$$

which implies that

$$\frac{1}{k} \sum_{j=1}^k \|c^j - \hat{c}^j\|_2^2 \leq \frac{1}{400t^2}$$

by rearranging. By averaging, at least $(1 - 1/100t^2)k$ of the $k$ centers $j \in [k]$ satisfy $\|c^j - \hat{c}^j\|_2^2 \leq 1/4$. $\quad\square$

Note that Lemma C.7 only allows us to characterize the behavior of $(1 - 1/100t^2)k$ many cluster centers, which still allows for the possibility that the remaining $k/100t^2$ centers are able to fit many points with low cost. The following lemmas show that this cannot happen.

**Lemma C.8.** *Consider a set of $k'$ centers $\hat{c}^j \in \mathbb{R}^d$ for $j \in [k']$. Let $\{Z^i\}_{i=1}^{n'}$ be $n' \geq M$ points such that $Z^i$ takes the value $t$ on coordinate $l^i \in [d]$, and furthermore, we have $\left| \{i \in [n'] : Z_l^i = t\} \right| \leq M$ for any $l \in [d]$. Then, the cost of any clustering of these $n'$ points with $k'$ clusters is at least*

$$n'\frac{d}{4} - n\frac{\log((k'+1)d) + 9}{2} + \frac{4}{5}t^2(n' - 10k' \cdot M)$$

*Proof.* We first lower bound the cost of the $k'$ centers by a "random" part of the cost and the "spike" part of the cost. For each $j \in [k']$, define the center $\bar{c}^j$ which is the center $\hat{c}^j$ with all entries greater than 1 set to 1.

Suppose that $Z^i$ is a point with some coordinate $l \in [d]$ such that $Z_l^i = t$. Note then that on the $l$th coordinate, we have that

$$(Z_l^i - \hat{c}_l^j)^2 \geq (Z_l^i - \hat{c}_l^j)^2 + (b^i - \bar{c}_l^j)^2 - 1$$

for some random bit $b^i \sim \{0, 1\}$. For all other coordinates $l \in [d]$, if $\hat{c}^j_l > 1$, then we lower bound the cost on the $l$th coordinate by

$$(Z^i_l - \hat{c}^j_l)^2 \geq (Z^i_l - 1)^2 + (1 - \hat{c}^j_l)^2 = (Z^i_l - \bar{c}^j_l)^2 + (1 - \hat{c}^j_l)^2$$

while if $\hat{c}^j_l \leq 1$, then we simply write the cost as $(Z^i_l - \hat{c}^j_l)^2 = (Z^i_l - \bar{c}^j_l)^2$. Note then that the cost lower bounds derived above can be grouped into a cost corresponding to a clustering cost of random bit vectors with centers $\bar{c}^j \in \mathbb{R}^d$, and everything else.

We will first lower bound the latter costs. Note that these costs are given by $(t - \hat{c}^j_l)^2 - 1$ for the coordinate $l \in [d]$ such that $Z^i_l = t$ and $(\hat{c}^j_l - 1)^2$ for the coordinates $l \in [d]$ such that $\hat{c}^j_l > 1$. In fact, we can note that this is just one less than the $\ell_2$ distance between $\hat{c}^j$ and the vector $(1, 1, \ldots, 1, t, 1, \ldots, 1)$, i.e., the all ones vector with $t$ in the $l$th position, since we can WLOG threshold all entries of $\hat{c}^j$ less than 1 to be exactly 1. Note that this cost is minimized when there are $n'/M$ different indices $l \in [d]$, each which has $\left|\{i \in [n'] : Z^i_l = t\}\right| = M$, and when all vectors $Z^i$ with the same coordinate $l$ for $Z^i_l = t$ are clustered to the same center (see, e.g., Fernandez et al. [2019]). For each $l \in [d]$, denote by $G^{(l)}$ the set $\{i \in [n'] : Z^i_l = t\}$. Then, there are at most $10k'$ indices $l \in [d]$ that belong to clusters consisting of at most 10 groups $G^{(l)}$. All other indices $l \in [d]$ belong to clusters that consist of at least 10 groups $G^{(l)}$, and thus the center of this cluster has coordinates with magnitude at most $t/10$. Thus, for at least $n' - 10k' \cdot M$ points, the cost is at least $(t - t/10)^2 = (9/10)^2 t^2 \geq (4/5)t^2$.

Next, we lower bound the cost of clustering the random bit vectors by $\bar{c}^j$. By Lemma C.5, the total cost of any clustering of $n$ random points with $k' + 1$ clusters must be at least

$$n\left(\frac{d}{4} - \frac{\log((k'+1)d) + 9}{2}\right).$$

One way to cluster these $n$ random points is to first cluster $n'$ points using $k'$ clusters, and then cluster all the remaining $n - n'$ points with the fixed center given by the vector with all $1/2$s, which gives a cost of $d/4$ for any point. Then by the above cost lower bound, it follows that the cost of the clustering of the $n'$ points using the $k'$ clusters must be at least

$$n\left(\frac{d}{4} - \frac{\log((k'+1)d) + 9}{2}\right) - (n - n')\frac{d}{4} = n'\frac{d}{4} - n\frac{\log((k'+1)d) + 9}{2}.$$

$\square$

### C.2.4 Reduction from Point Assignment

Finally, we obtain an information complexity lower bound for the $k$-means clustering problem, by a reduction from the point assignment problem of Lemma C.3.

**Theorem C.1.** *Let $t = \max\{2000, 80\sqrt{\log(kd) + 10} + 2\}$. Let $\{Z^i\}_{i=1}^n$ be drawn i.i.d. from the distribution of Definition C.2, with $\alpha = 1/100t^2$. Consider the input instance given by these points, together with the planted centers as specified in Lemma C.7. Suppose that $\hat{c}^j \in \mathbb{R}^d$ for $j \in [k]$ are centers that achieve a $(1 + \epsilon)$ approximation, for $\epsilon = (\log(kd) + 10)/(d + (t - 1)^2) = \tilde{O}(1/d)$. Suppose that we assign $e_l$ to $Z^i$ whenever $Z^i$ is clustered to the center $\hat{c}^j$ that has largest entry in the $l$th coordinate for $l \in [d]$. Then, this solves the point assignment problem (Definition C.1) for at least $(399/400)n$ of the $Z^i$ for $i \in [n]$. Hence, solving $k$-means clustering up to $(1 + \epsilon)$ accuracy on this instance requires $\Omega(nd/t^2) = \tilde{\Omega}(nd) = \tilde{\Omega}(n/\epsilon)$ bits of communication.*

*Proof.* Let $\{\hat{c}^j\}_{j=1}^k$ be a clustering achieving a $(1 + \epsilon)$ approximation. We will show that we must have at most $n/400$ incorrect classifications of the points $Z^i$.

We first introduce some notation. For each $j \in [k]$, we let $G^{(j)} \subseteq [n]$ denote the subset of points $i \in [n]$ such that $Z^i_j = t$, and we let $G^{(0)} := [n] \setminus \bigcup_{j \in [k]} G^{(j)}$ denote the set of points such that $\|Z^i\|_\infty \leq 1$. Note then that $G^{(0)}$ corresponds to the set of points with $C = (0, 0, \ldots, 0)$ for $C$ defined in Definition C.2, and thus has size $\mathbb{E}\left|G^{(0)}\right| = \alpha n$ in expectation and size $\Theta(\alpha n)$ with

probability at least $1 - \delta$ by Chernoff bounds. We will also define $\bar{c}^j$ for each $j \in [k]$ to be the center $\hat{c}^j$ with any entry larger than 1 set to be equal to 1.

By Lemma C.7, there is a subset $S \subseteq [k]$ of size at least $|S| \geq (1 - 1/100t^2)k$ such that $\|c^j - \hat{c}^j\|_2^2 \leq 1/4$. We make use of this fact later, and first bound the cost of points that can be clustered by the remaining at most $k' = |[k] \setminus S| \leq k/100t^2$ centers. Note that by Chernoff bounds and a union bound, we have that $\left|\{i \in [n] : Z_l^i = t\}\right| \leq 2n/k$ for every $l \in [n]$. Then by Lemma C.8, if there are $n'$ points clustered by these $k'$ centers, then the cost is at least

$$n'\frac{d}{4} - n\frac{\log(kd) + 9}{2} + \frac{4}{5}t^2\left(n' - 10\frac{k}{100t^2}\frac{2n}{k}\right) \geq n'\left(\frac{d}{4} + \frac{4}{5}t^2\right) - n\frac{\log(kd) + 10}{2} \quad (6)$$

Now let $j \in S$. We will bound the cost of the points $Z^i \in G^{(j)}$, as a function of the number of points that are clustered to some center $\hat{c}^{j'}$ for $j' \neq j$. Let $Z^i$ be a point clustered to some center $\hat{c}^{j'}$ for $j' \neq j$ and $j' \in S$ (recall that we have already handled the cost of clustering points to centers outside of $S$). Then, the cost on the $j$th coordinate is bounded below by

$$(Z_j^i - \hat{c}_j^{j'}) \geq \left(t - \frac{1}{2} - \|c^{j'} - \hat{c}^{j'}\|_\infty\right)^2 \geq (t - 1)^2.$$

On the other hand, if the assigned center is correct, i.e. $j' = j$, then the cost lower bound on the $j$th coordinate is

$$(Z_j^i - \hat{c}_j^{j'}) \geq \left(t - \frac{t+1}{2} - \|c^{j'} - \hat{c}^{j'}\|_\infty\right)^2 \geq (t - 2)^2/4.$$

Thus, each incorrectly classified point pay an additional cost $(t - 1)^2 - (t - 2)^2/4 \geq (t - 2)^2/2$ on the $j$th coordinate. We will later lower bound the cost of the rest of the coordinates via Lemma C.5.

In the last remaining cases of $i \in G^{(0)}$ and $i \in G^{(j)}$ for $j \notin S$, we will only be able to lower bound the cost by the cost of the random coordinates via Lemma C.5, but not by the additional $(t - 2)^2/4$ term on the $j$th coordinate. This will be fine, as there are only roughly $n/t^2$ such points, since $\left|G^{(0)}\right| \leq 2\alpha n = n/50t^2$ and $|[k] \setminus S| \leq (1/100t^2)k$ so

$$\left|\bigcup_{j \in [k] \setminus S} G^{(j)}\right| \leq \frac{k}{100t^2}\frac{2n}{k} \leq \frac{n}{50t^2}.$$

Thus, at least $n - n' - (n/50t^2 + n/50t^2)$ points will incur a cost of $(t - 2)^2/4$, for a cost contribution of

$$\frac{(t - 2)^2}{4}\left(n - n' - (n/50t^2 + n/50t^2)\right) = (n - n')\frac{(t - 2)^2}{4} - \frac{n}{100}$$

Finally, we bring all the above calculations together. Suppose that there are $b$ points $Z^i$ that belong to $G^{(j)}$ for some $j \in S$, but are clustered to some other $\hat{c}^{j'}$ for $j' \in S$. First, the cost of the points that are clustered to some center not in $S$ is given in (6). Next, the cost of clustering the random coordinates of all other points is similarly bounded below by Lemma C.5 by

$$(n - n')\frac{d}{4} - n\frac{\log(kd) + 9}{2}.$$

Thus, altogether, the cost is bounded below by

$$b\frac{(t - 2)^2}{2} + (n - n')\left(\frac{d}{4} + \frac{(t - 2)^2}{4}\right) + n'\left(\frac{d}{4} + \frac{4}{5}t^2\right) - n(\log(kd) + 10)$$

$$\geq b\frac{(t - 2)^2}{2} + \frac{nd}{4} + n\frac{(t - 2)^2}{4} + n'\frac{(t - 2)^2}{2} - n(\log(kd) + 10)$$

Then, if $b$ or $n'$ are greater than $n/800$, then this cost is at least

$$\frac{n}{800}\frac{(t - 2)^2}{2} + \frac{nd}{4} + n\frac{(t - 2)^2}{4} - n(\log(kd) + 10)$$

For $t \geq 2000$, we have that
$$\frac{1}{2}\frac{n}{800}\frac{(t-2)^2}{2} \geq \frac{n}{4} \cdot 2t$$
and for $t \geq 80\sqrt{\log(kd)+10}+2$, we have that
$$\frac{1}{2}\frac{n}{800}\frac{(t-2)^2}{2} \geq 2n(\log(kd)+10)$$
and thus if both of these hold, then the cost is at least
$$\frac{nd}{4} + n\frac{(t-1)^2}{4} + n(\log(kd)+10).$$

Thus, by our choice of $\epsilon$, this fails to be a $(1+\epsilon)$-approximate solution, and thus we must have that $b$ and $n'$ are both at most $n/800$. Thus, the algorithm can incorrectly classify at most $n/400$ points. $\qquad\square$

## D   Missing Proofs from Section 4

### D.1   Proof of Theorem 4.2

*Proof of Theorem 4.2.* Let $d = 2\lceil 1/\epsilon^2\rceil$ and let $X = \{X^i\}_{i=1}^n \subseteq \{0,1\}^{d/2}$ be a collection of $n$ uniformly random bit vectors, each with $d/2$ coordinates. Then for each $i \in [n]$, we form a vector $a^i \in \mathbb{R}^d$ by setting the $(2j-1)$th and $2j$th coordinates to be

$$(a_{2j-1}^i, a_{2j}^i) = \begin{cases} (0,1) & \text{if } X_j^i = 0 \\ (1,0) & \text{if } X_j^i = 1 \end{cases}$$

Fix any $j \in [d/2]$, and suppose that we query the cost of two centers given by the vectors $c^1 = \sqrt{d} \cdot e_{2j-1}$ and $c^2 = \sqrt{d} \cdot e_{2j}$. Then, the center cost query data structure must output a partition $C^1, C^2 \subseteq [n]$ such that

$$\sum_{i \in C^1} \|a^i - c^1\|_2^2 + \sum_{i \in C^2} \|a^i - c^2\|_2^2 \leq (1 + \epsilon/15)\operatorname{cost}(c^1, c^2).$$

We claim that the partition must assign all but at most $n/10$ of the $a^i$ to its closest center. Note that this implies the theorem. Indeed, given the center cost query data structure $M$, we can reconstruct a bits $X'$ which agrees with $X$ on all but at most $(n/10)(d/2) = nd/20$ bits, so

$$\begin{aligned} \mathsf{H}(M) &\geq \mathsf{H}(M) - \mathsf{H}(M \mid X) \\ &= \mathsf{I}(M; X) \\ &\geq \mathsf{I}(X'; X) & \text{data processing inequality} \\ &= \mathsf{H}(X) - \mathsf{H}(X \mid X') \\ &\geq \frac{nd}{2} - \frac{nd}{20} = \Omega(nd). \end{aligned}$$

Then, $M$ must use at least $\Omega(nd)$ bits to describe, since the number of bits of a message upper bounds the entropy of a random variable.

Note first that the cost of this query on any vector is at least

$$(\sqrt{d}-1)^2 \geq (1 - 1/\sqrt{d})^2 d \geq (1 - 2/\sqrt{d})d \geq d/2$$

and at most

$$\|a^i - c^1\|_2^2 \leq 2\|a^i\|_2^2 + 2\|c^1\|_2^2 = 3d.$$

Thus, the total error that the partition can incur is at most

$$\sum_{i \in C^1} \|a^i - c^1\|_2^2 + \sum_{i \in C^2} \|a^i - c^2\|_2^2 - \operatorname{cost}(c^1, c^2) \leq \epsilon\operatorname{cost}(c^1, c^2) \leq 3 \cdot \frac{\epsilon}{15}nd \leq \frac{\sqrt{d}}{5}n$$

By averaging over the $n$ vectors, there can be at most $n/10$ indices $i \in [n]$ such that $a^i$ is assigned to a cluster with center $c \in \{c^1, c^2\}$ with

$$\|a^i - c\|_2^2 - \min\{\|a^i - c^1\|_2^2, \|a^i - c^2\|_2^2\} \geq 2\sqrt{d}$$

Now consider a single vector $a^i$, and say that $(a^i_{2j-1}, a^i_{2j}) = (0, 1)$. Note then that the difference between the cost of assigning this vector to $c^1$ versus the cost of assigning this vector to $c^2$ is at least

$$(\sqrt{d})^2 + 1^2 - (\sqrt{d} - 1)^2 \geq 2\sqrt{d}.$$

Thus, there are at most $n/10$ vectors that can be assigned to the incorrect center. $\qquad\square$

## D.2 Proof of Theorem 4.3

*Proof of Theorem 4.3.* The proof is by a reduction from set disjointness Razborov [1990]. Suppose that Alice and Bob are two players who hold an instance of set disjointness, that is, Alice has a subset $A \subseteq [n]$ and Bob has a subset $B \subseteq [n]$, and they must determine whether $A \cap B$ is empty or not by sending each other messages in any number of rounds. It is known that any randomized algorithm solving this task with probability at least $2/3$ requires $\Omega(n)$ bits of communication Razborov [1990].

Suppose that there is a randomized turnstile streaming algorithm $\mathcal{A}$ which can output a relative error approximation to the $k$ means clustering cost with probability at least $2/3$ while using $r$ passes and space at most $M$. Then, we claim that Alice and Bob can use this algorithm to solve set disjointness in $2rM$ bits of communication, which implies that $M = \Omega(n/r)$. To do this, Alice first runs the algorithm $\mathcal{A}$ on the input stream which updates $\mathbf{A}_{i,1} \leftarrow \mathbf{A}_{i,1} + 1$ for every $i \in A$. Then, Alice sends the memory state of $\mathcal{A}$, which is at most $M$ bits, to Bob. Bob then continues to run the algorithm $\mathcal{A}$ by updating running it on the stream which updates $\mathbf{A}_{i,1} \leftarrow \mathbf{A}_{i,1} + 1$ for every $i \in B$. Finally, Bob also adds two dummy coordinates which has entries $0$ and $1$ each. Bob can then send the memory state back to Alice, which again is at most $M$ bits. This can be repeated for $r$ passes, for a total of $2rM$ bits of communication.

We now show that given an estimate $c$ satisfying (2), Alice and Bob can determine whether $A \cap B$ is empty or not. If $A \cap B$ is empty, then note that all rows of $\mathbf{A}$ are either $0$ or $1$, so the $k$-means clustering cost for $k = 2$ is $0$ and thus $c$ must be $0$. On the other hand, if $A \cap B$ is nonempty, then there is at least one row of $\mathbf{A}$ that is $2$ as well as a $0$ and a $1$ from the two dummy coordinates added by Bob, so the cost is strictly positive. Thus, $c$ must be strictly positive in this case. $\qquad\square$

## D.3 Proof of Theorem 4.4

*Proof of Theorem 4.4.* Our proof for this result roughly follows our proof of Theorem 4.3, so we only point out the important changes. We again let Alice and Bob have subsets $A \subseteq [n]$ and $B \subseteq [n]$, respectively. However, for this reduction, we construct our input instance $\mathbf{A}$ to be $(2n + 3) \times 1$. First, Alice inserts her items $i \in A$ from $A$ in two coordinates, updating $\mathbf{A}_{2i,1} \leftarrow \mathbf{A}_{2i,1} + 1$ and $\mathbf{A}_{2i+1,1} \leftarrow \mathbf{A}_{2i+1,1} + 1$ for every $i \in A$. Similarly, Bob updates $\mathbf{A}$ in the two coordinates $\mathbf{A}_{2i,1} \leftarrow \mathbf{A}_{2i,1} + 1$ and $\mathbf{A}_{2i+1,1} \leftarrow \mathbf{A}_{2i+1,1} + 1$ for every $i \in B$. Finally, Bob inserts three dummy coordinates which has entries $0$, $1$, and $3$.

We now claim that an approximate set of centers $\tilde{\mathbf{D}}$ can distinguish the cases between $A \cap B$ empty and $A \cap B$ nonempty. In the former case, the set of centers output by the $k$-means clustering algorithm must be $\{0, 1, 3\}$, since this is the unique solution with a cost of $0$. On the other hand, if $A \cap B$ is nonempty, then we claim that the $k$-means clustering algorithm cannot output $\{0, 1, 3\}$. Indeed, in this case, the cost of this solution is at least $2$ since there are at least two coordinates whose value is $2$. On the other hand, the solution of $\{0, 1, 2\}$ has a cost of $1$, since there is only a single dummy coordinate of $3$ that does not intersect exactly with these centers. $\qquad\square$

## D.4 Proof of Theorem 4.5

We will need the following sensitivity sampling theorem:

**Theorem D.1** (Sensitivity sampling, Feldman and Langberg [2011], Braverman et al. [2016], Woodruff and Yasuda [2023]). *Let*

$$\tilde{\sigma}_i \geq \sup_{c^1, c^2, \ldots, c^k \in \mathbb{R}^d} \frac{\min_{j=1}^k \|a^i - c^j\|_2^2}{\sum_{i'=1}^n \min_{j=1}^k \|a^{i'} - c^j\|_2^2}$$

and $\tilde{\mathfrak{S}} := \sum_{i=1}^{n} \tilde{\sigma}_i$. *Suppose that for each $i \in [n]$, $a^i$ is sampled independently with probability $p_i := \min\{1, \tilde{O}(\tilde{\sigma}_i kd/\epsilon^2)\}$, with an associated weight $w_i = 1/p_i$ if $i$ is sampled and $0$ otherwise. Then, for every $c^1, c^2, \ldots, c^k \in \mathbb{R}^d$, we have that*

$$\sum_{i=1}^{n} \min_{j=1}^{k} \|a^i - c^j\|_2^2 = (1 \pm \epsilon) \sum_{i=1}^{n} w_i \min_{j=1}^{k} \|a^i - c^j\|_2^2.$$

We then obtain the following result:

*Proof of Theorem 4.5.* Note that if a dataset has sensitivities bounded by $\alpha$, then a uniformly random sample of size $\tilde{O}(\alpha nkd/\epsilon^2)$ is a sample as given in Theorem D.1. Thus, approximately optimal centers $\hat{c}^1, \hat{c}^2, \ldots, \hat{c}^k \in \mathbb{R}^d$ are approximately optimal centers for the entire dataset. These centers can be found using just

$$\tilde{O}((\alpha nkd/\epsilon^2)/\epsilon^2 + dk/\epsilon) = \tilde{O}(\alpha nkd/\epsilon^4 + dk/\epsilon)$$

bits of space, using our turnstile streaming $k$ means clustering result (Theorem 3.2). Furthermore, because the input stream is a random order stream, these approximately optimal centers $\hat{c}^1, \hat{c}^2, \ldots, \hat{c}^k$ can be obtained after seeing the first $\tilde{O}(\alpha nkd/\epsilon^2)$ elements of the stream. With approximately optimal centers in hand, note that the rest of the $n - \tilde{O}(\alpha nkd/\epsilon^2)$ points can be assigned on the fly, and thus space complexity is just an additional $O(n \log k)$ bits. $\qquad\square$