# OpenReview forum: "Sketching Algorithms for Sparse Dictionary Learning: PTAS and Turnstile Streaming"
_NeurIPS.cc/2023/Conference — NeurIPS 2023 poster_

### Official Review · Reviewer_NUiP · 2023-07-02

**Soundness:** 4 excellent
**Presentation:** 3 good
**Contribution:** 3 good
**Rating:** 6
**Confidence:** 4

**Summary:**

The paper studies the sparse dictionary learning and k-means clustering problems, using tools from sketching. Various results are obtained under different settings and assumptions.

The first part of the paper considers lower bounds in the streaming setting. Here, the main technical result, which is a lower bound for k-means, uses a reduction from the communication complexity of the multiparty set-intersection problem. This is a nice set of results that translate prior ideas to k-means clustering problems. A new upper bound is also established under certain input sensitivity assumptions in the random order model.

The second part of the paper gives approximation schemes for both problems. The key technical ingredient is the use of dimensionality reduction. Here, the use of dimensionality reduction follows essentially from prior work on projective clustering.

The final part of the paper considers space complexity in the turnstile model. At the high level, sketching tools are used to discretize the appropriate matrix optimization problem, followed by brute force.

**Strengths:**

The paper studies two fundamental important problems. The work introduces several ideas from sketching in this setting.

The authors appear to be very well versed in the related literature.

The paper is well-written and the presentation is quite accessible.


**Weaknesses:**

In the statements of several of the Theorems, the running time is not given (either at all, or not with enough precision). This makes me think that the algorithms are probably not very practical.

The section on the turnstile model is labeled as "space complexity". I find this confusing. Your algorithm has some space complexity and some time complexity, and it would be best to state both clearly.


**Questions:**

Can you provide any bounds on the exponent in the poly(n) term in Theorem 3.2?

What's the running time of the algorithm in Theorem 4.3?

Can you comment on possible applicability of your algorithms? Is there a bottleneck in performance?

Can you briefly discuss whether your results on k-means imply anything for k-center or k-median? Your lower bound constructions seem general enough that perhaps other problems can be addressed too.


**Limitations:**

The authors have adequately addressed the limitations of their work.

---

> ### Author Rebuttal · Authors · 2023-08-10
>
> >In the statements of several of the Theorems, the running time is not given (either at all, or not with enough precision). This makes me think that the algorithms are probably not very practical.
>
> We will add more precise time complexity results to our upper bounds, so that our work is easier to compare to in future work. Regarding the practicality of our methods, please see our later response regarding the "bottleneck" of our methods.
>
> > The section on the turnstile model is labeled as "space complexity". I find this confusing. Your algorithm has some space complexity and some time complexity, and it would be best to state both clearly.
>
> We will add time complexity bounds to the statements of Theorem 4.1 and 4.2 as well as rename the section to "Turnstile Streaming Algorithms".
>
> > Can you provide any bounds on the exponent in the $\operatorname{poly}(n)$ term in Theorem 3.2?
>
> Ignoring lower order terms, the complexity of Theorem 3.2 will be $\exp((8k^{3r}b \log d)^{O(k^{r+1})}\log n)$.
>
> > What's the running time of the algorithm in Theorem 4.3?
>
> The dominant time complexity term for Algorithm 4.3 would be $n^{\tilde{O}(k^2/\epsilon)}$, since the cost with respect to $d$ is amortized over the turnstile updates.
>
> > Can you comment on possible applicability of your algorithms? Is there a bottleneck in performance?
>
> The main bottleneck generally comes from the fact that, once we reduce the dimension of a problem, solving the reduced problem has a high computational complexity in terms of parameters such as $k$ and $r$. This likely cannot be avoided in the worst case, since both $k$-means and sparse dictionary learning are NP-Hard. However, there has been extensive work studying heuristic algorithms and algorithms leveraging additional statistical/structural assumptions for these problems. Our dimensionality reduction approaches could potentially be paired with these methods to achieve more practically efficient algorithms.
>
> > Can you briefly discuss whether your results on k-means imply anything for k-center or k-median? Your lower bound constructions seem general enough that perhaps other problems can be addressed too.
>
> Thank you for pointing this out! Indeed, we expect our lower bound argument to hold for Euclidean $k$-median clustering, and more generally, $(k,p)$-clustering for $p \leq 2$. For $p > 2$ (including $k$-center clustering), our lower bounds may also generalize to this setting, but we may also expect stronger lower bounds since $\ell_p$ norm estimation in $d$ dimensions requires $\operatorname{poly}(d)$ bits of space for $p>2$. In general, generalizing our results from Euclidean metrics to $\ell_p$ metrics is an interesting question.

---

### Official Review · Reviewer_MGS2 · 2023-07-03

**Soundness:** 3 good
**Presentation:** 2 fair
**Contribution:** 4 excellent
**Rating:** 6
**Confidence:** 4

**Summary:**

This theoretical paper discusses algorithms and lower bounds regarding time and space complexity for two interesting machine learning problems:

1) Euclidean k-means clustering

2) Sparse dictionary learning


I briefly summarize the results based on the order that they are presented in the main paper (which does not fully agree with the order that is presented in the introduction):

1. Space complexity **lower bounds** for approximate k-means in the **(a)** turnstile streaming, **(b)** row-arrival streaming

2. Upper bound for a special case of k-means, where the so-called sensitivities are bounded, which goes beyond the aforementioned lower bounds.

3. PTAS for dictionary learning and k-means based on random dimension reduction.

4. Space complexity **upper bounds** for the two problems in the turnstile streaming model.

All the results are supported with detailed proofs. Admittedly I only briefly checked very few of the proofs due to the limited time, but from the general impression of the paper I expect them to be robust.

**Strengths:**

The major strengths that I can highlight are the following:

1) The problems that are being studied are related to important machine learning problems that have drawn a lot of attention in the past

2) The results that are presented reveal many interesting insights for these two problems, from an algorithmic perspective, and the algorithms communities can benefit from such a detailed analysis.

3) Many results are presented and heavily supported with theoretical analysis (this is mostly a strength, but the amount of work makes it hard to thoroughly review in the limited time that is available)

4) It is evident that there has been a lot of work preparing the paper and it seems that the majority the main claims are robust.

**Weaknesses:**

The major weakness of the paper that I can mention, unfortunately, is the way that it is written...!

1) There is no clear-cut summary of the results in the introduction. There are many interesting results and improvements, but it takes significant time to identify them. E.g., the order in which the results are presented in the introduction does not agree with the order that the results are presented in the sections thereafter. The first mention of contributions is PTAS, but this only appears in Section 3. Section 2, which precedes PTAS, is about space complexity lower bounds. I spent a lot of time trying to locate results and connect them with each other, which I could have spent in verifying proofs.

2) There are many results in the 20 pages of additional content seem to be new, non-trivial, and seem to be crucial parts of the paper, e.g. Algorithms 1 and 2, the polysolver, .... They should be part of the main paper, it is not ideal that they remain hidden in the Appendix.

3) There are few things that I think need to be clarified / missing definitions, See "Questions".

**Note**: Items 1. and 2. did not affect my score.


**Questions:**

I have the following questions for the authors that could help me understand better the paper and potentially modify the score.

**Questions**:

1) Introduction, page 3: It is mentioned that a Gaussian JL sketch only need $O(ε^{-2}n)$ bits of space, could you explain a bit more? Maybe another JL sketch is required? (e.g. Rademacher,...)

2) Theorems 2.3 and Theorem 2.4 mention "a constant number of passes...". Does the model support more than 1 passes over the stream? This is not mentioned in Definition 1.3 of a turnstile stream.

3) Theorem 3.1: why is it needed that rank(A)=Ω(poly(k/ε))? How is it ensured later on? (an assumption of d=poly(k/ε) is made for d, not for rank(A))

4) Theorem 3.2: $b$ is not defined. Also vec(A), which is first used in page 9, is not defined

5) Abstract: I am not sure why the bounded sensitivity assumption is natural. Could you explain a bit more?

6) Line 342: S seems to also depend on ε.

7) Line 26: $r$-sparse is not defined. It might be confusing to someone who is not familiar with the term.

8) I am not sure I understand the derivation of the PTAS (e.g. Section 3.4). The polysolver in the appendix seems to run in time exponential to the size of the input problem. From what I understood, this should not be a problem because we already reduce the matrix problem size with a sketch to something like $O(k log k)$, and if $k$ is fixed then $exp(k)$ is fine for a PTAS. Is this more or less correct? Could you provide a brief description that could help me understand the proofs of that section better?


**Typos / Other suggestions**:

1) Theorem 2.1: "k-maens"

2) Line 64: Should it be "efficient" instead of "inefficient"?

3) The title of Section 4 is a bit too generic, it took me a while to understand why it is any different from section 2, which also describes space complexity for turnstile.

**Main recommendation**: I think that the structure of the paper can be significantly improved with not too much effort, and it would substantially improve the overall impression of the paper. I.e., there should be a clear summary of the new (and interesting) results that can guide the reader to locate them easier. I really think the best fit for this (large) paper is to submit a revised full-version to a ML or TCS journal, where the reviewers can take the appropriate amount of time to provide insightful feedback, and the suggestions could be incorporated in a revision. In any case, this is up to the authors.


**Limitations:**

I cannot see any potential negative societal impact.

---

> ### Author Rebuttal · Authors · 2023-08-10
>
> > There is no clear-cut summary of the results in the introduction. There are many interesting results and improvements, but it takes significant time to identify them. E.g., the order in which the results are presented in the introduction does not agree with the order that the results are presented in the sections thereafter. The first mention of contributions is PTAS, but this only appears in Section 3. Section 2, which precedes PTAS, is about space complexity lower bounds. I spent a lot of time trying to locate results and connect them with each other, which I could have spent in verifying proofs.
>
> Thank you for pointing out that the order of sections makes our work harder to read, and we apologize for the inconveniences caused by this. We have reordered the sections of the main body of our work to follow the flow of the introduction.
>
> > Introduction, page 3: It is mentioned that a Gaussian JL sketch only need $O(\varepsilon^{-2}n)$ bits of space, could you explain a bit more? Maybe another JL sketch is required? (e.g. Rademacher,...)
>
> You are correct that exact instances of Gaussian random variables cannot be represented in finite space. However, Gaussian matrices truncated to finite bit complexity will suffice. We will avoid referring to $\mathbf{G}$ as a Gaussian matrix and instead we can refer the reader (among other places) to Definition 1.2 in [Makarychev et. al., 2020].
>
> > Theorems 2.3 and Theorem 2.4 mention "a constant number of passes...". Does the model support more than 1 passes over the stream? This is not mentioned in Definition 1.3 of a turnstile stream.
>
> While turnstile streams can indeed be defined for more than one pass over the stream, our algorithms only need one pass, while our lower bounds work against a constant number of passes. We will clarify that our lower bounds work against this broader class of algorithms compared to our upper bounds in our revision.
>
> > Theorem 3.1: why is it needed that rank(A)=Ω(poly(k/ε))? How is it ensured later on? (an assumption of d=poly(k/ε) is made for d, not for rank(A))
>
> Good point. We do not need this assumption since, if $rank(\mathbf A) < s$, then we can just directly reduce the dimension of the problem using SVD. We will change the theorem to remove this assumption. We do not need to change any later theorems, as we did not use this rank assumption anywhere else.
>
> > Theorem 3.2: $b$ is not defined. Also vec(A), which is first used in page 9, is not defined.
>
> Thank you for pointing this out. We write $b$ to denote the bit complexity of the input matrix $\mathbf A$, that is, each entry of $\mathbf A$ can be represented by $b$ bits. This is stated in Theorem D.1 of our initial draft, and we will move this definition to the main text in our revision. We have also clarified that vec(A) refers to the $nd$-dimensional vector obtained by flattening $\mathbf A$.
>
> > Abstract: I am not sure why the bounded sensitivity assumption is natural. Could you explain a bit more?
>
> The sensitivity quantity captures how sensitive the objective function is with respect to a given point $i\in[n]$, and is defined as the largest fraction of the objective function captured by the training example $i\in[n]$, ranging over all centers $c^1, c^2, \dots c^k\in\mathbb R^d$. The bounded sensitivity assumption states that there are no points that can take up a significant fraction of the objective function, and can also be interpreted as a way to formalize a ``well-clustered'' instance. In particular, this is one assumption which makes uniform sampling a good algorithm for sampling a small representative subset of examples. We have given a more in-depth discussion of this assumption in our revision.
>
> > Line 342: S seems to also depend on $\epsilon$.
>
> Thank you for catching this, we have fixed this in our revision.
>
> > Line 26: $r$-sparse is not defined. It might be confusing to someone who is not familiar with the term.
>
> We have clarified that an $r$-sparse linear combination is a linear combination of at most $r$ vectors.
>
> > I am not sure I understand the derivation of the PTAS (e.g. Section 3.4). The polysolver in the appendix seems to run in time exponential in the size of the input problem. From what I understood, this should not be a problem because we already reduce the matrix problem size with a sketch to something like $O(k\log k)$, and if is fixed $\exp(k)$ then is fine for a PTAS. Is this more or less correct? Could you provide a brief description that could help me understand the proofs of that section better?
>
> Yes, what you have described is indeed correct. Although the stated time complexity of PolySolver is $2^{O(nr + kd)}$ for input matrix $\mathbf{A}$, we never call the PolySolver on the full matrix, but instead call it on $\mathbf{W}\mathbf{A}$, which has a smaller dimension that is only logarithmic in $n$. To briefly give the intuition behind the proof of our sparse dictionary learning PTAS, we first assume $d = \operatorname{poly}(k/\epsilon)$ (this is justified since we give a way to efficiently reduce to this case in Theorem 3.1). We then reduce the size of $\mathbf{A}$ to be logarithmic in $n$ by applying a projective clustering coreset construction. We then show that using existing work on polynomial system solvers, we can solve this smaller problem in $\operatorname{poly}(n)$ time. The technical difficulty comes from rigorously interfacing the projective clustering bounds, polynomial system solver complexity, and the specific constraints of the sparse dictionary learning problem, and this is what causes the technical complexity in the proof despite the intuitive simplicity of the approach.

---

> > ### Comment · Reviewer_MGS2 · 2023-08-10
> >
> > I would like to thank the authors for their responses and for taking all the reviews into consideration. I was not able to find the revised manuscript (the "revisions" page is empty). Could you point me to it?

---

> > > ### Author Response · Authors · 2023-08-10
> > >
> > > While we did our best to describe in detail the revisions that we plan to make, per NeurIPS policy, no revisions are allowed until the camera-ready stage (https://neurips.cc/Conferences/2023/PaperInformation/NeurIPS-FAQ), and we refrained from uploading a revised paper.

---

> > > > ### Comment · Reviewer_MGS2 · 2023-08-10
> > > >
> > > > Ok thank you for the quick response, I will evaluate as best as possible from the existing content (I can certainly say the rating will not go down)

---

> > > > > ### Comment · Reviewer_MGS2 · 2023-08-14
> > > > >
> > > > > After taking into account all the reviews and discussions I have decided to raise my rating from 5 to 6. From my perspective, the structure and the writing indicate that the paper should have been passed over 1 or 2 more times before submission. However, the responses of the authors and the general impression that I got from the analysis and the technical content suggest that the authors are rather confident with their results. I was not able to proof-read everything in the (big) paper, but it is enough to give the benefit of doubt, and I trust that the revised manuscript will deserve to be published.

---

### Official Review · Reviewer_rhYv · 2023-07-04

**Soundness:** 4 excellent
**Presentation:** 4 excellent
**Contribution:** 3 good
**Rating:** 7
**Confidence:** 3

**Summary:**

The paper considers the well studied $k$-means clustering problem and the $r$-sparse dictionary learning problem. The paper has multiple contributions:

(1) It presents a new approach for obtaining a PTAS for $k$-means clustering which matches the time complexity of previous algorithms for the problems. This approach generalizes to give the first PTAS for the sparse dictionary problem.

(2) Within turnstile streaming algorithms, they consider the setting where the algorithm has to output both the assignments to the clusters/dictionaries as well as the cluster centers or dictionary elements. Previous work, even in the case of the simpler $k$-means have either focused on one or the other so this is a more challenging setting. Omitting logarithmic factors, the paper provides an $O(nr/\varepsilon^2+dk/\varepsilon)$ space algorithm for the $r$-sparse dictionary learning problem with dictionaries of size $k$ and an $O(n/\varepsilon^2+dk/\varepsilon)$ space algorithm for the $k$-center problem. They also present an $O(n)$ space bounded algorithm when the points are inserted in a random order. On the lower bound side, they present an $\Omega(n/\varepsilon+dk/\varepsilon)$ space bound for $k$-means clustering as well as an $\Omega(n/\varepsilon^2)$ bound for algorithms that can estimate the cost for a fixed set of candidate centers.

Technically most interesting seems to be the former lower bound for $k$-means clustering which is via a reduction from the multi-party set disjointness problem.

**Strengths:**

I found the paper to be quite strong. It is indeed surprising that the setting where both the assignments and the centers/dictionaries must both be output have only received limited attention. I found the ideas for the lower bound via multi-party set disjointness interesting (they are sketched well in the first 9 pages) and I als think it is nice that they design a PTAS for dictionary learning. I went through a few of the proofs in the appendices but far from everything, so I cannot vouch for correctness. However, the paper is well written and the proofs seem clear. Given that both the $k$-center problem and sparse dictionary learning is of interest to a good chunk of the NeurIPS community, I think the paper should be accepted.

**Weaknesses:**

It seems that some of the algorithms proposed by the paper might be implementable and it would be nice to see some experiments on their performance.

**Questions:**

l64: Should "inefficient" be "efficient"?

l104-105: you should probably say "after applying $G$" somewhere.

Definition 2.3. This definition is strange. Saying that the algorithm outputs an $\varepsilon$-approximation does not have anything to do with the rows arriving one at a time. It seems that it should be a definition of the model and not say anything about the approximation.

l236: Appending $k$ rows of what?

l238: "but a $k$ rows". Please check the writing.

l263: What is an indicator matrix?

l320: I am a bit confused why the dimensionality has been reduced to logarithmic in $n$. The new dimension seems to only depend on $k$ and $r$.

l331: Has $b$ been introduced?

Lemma 3.1: Please check the statement. Are the order of quantifiers correct? Should the bound on $s$ be moved further back?

l360: I think $S$ is $m\times n$. Same for l380.


**Limitations:**

None as far as I can tell

---

> ### Author Rebuttal · Authors · 2023-08-10
>
> > It seems that some of the algorithms proposed by the paper might be implementable and it would be nice to see some experiments on their performance.
>
> While we agree that experimental inquiry on these streaming/sketching algorithms would be interesting, we believe they would be best situated in a work dedicated to the topic. Our results are focused on algorithmic techniques that will achieve the best worst-case complexity, which is related to, but not the same as optimizing for practical performance. Rather than directly apply our proposed algorithms, we believe our sketching methods to reduce the problem size could be paired with existing heuristic algorithms to efficiently solve the reduced problems in practice, at the cost of worst-case guarantees.
>
> > l64: Should "inefficient" be "efficient"?
>
> We have clarified this as "where even an inefficient algorithm will be tractable due to the smaller size of the instance".
>
> > l104-105: you should probably say "after applying $G$" somewhere.
>
> Done.
>
> > Definition 2.3. This definition is strange. Saying that the algorithm outputs an $\epsilon$-approximation does not have anything to do with the rows arriving one at a time. It seems that it should be a definition of the model and not say anything about the approximation.
>
> Indeed, this definition is intended to specify the model in which an input of $n$ vectors arrive one at a time. One could solve a variety of problems in this model, such as $\epsilon$-approximate $k$ means clustering, exact $k$ means clustering, or any other problem whose inputs are $n$ vectors in $d$ dimensions. We have clarified this in our revision.
>
> > l236: Appending $k$ rows of what?
>
> We have clarified this as follows: "The result of Woodruff (2014) constructs a distribution over $O(k/\epsilon)\times d$ matrices such that one can recover an arbitrary random bit among $\tilde\Omega(dk/\epsilon)$ random bits by appending a set of $k$ ``query'' rows and then computing a $(1+\epsilon)$-approximately optimal low rank approximation to the resulting matrix."
>
> > l238: "but a $k$ rows". Please check the writing.
>
> We have fixed this to be "all but $k$ rows".
>
> > l263: What is an indicator matrix?
>
> We have clarified this as follows: "let $\mathcal X$ be the set of matrices $\mathbf X\in\mathbb R^{n\times k}$ with standard basis vectors as rows".
>
> > l320: I am a bit confused why the dimensionality has been reduced to logarithmic in $n$. The new dimension seems to only depend on $k$ and $r$.
>
> We were not precise in saying the size and dimensionality were at most logarithmic in $n$, as this is all we need for the intuition of the argument to hold. However, you are correct that, more specifically, the dimension is reduced to be independent of $n$ and the size of the input is reduced to be logarithmic in $n$. We will improve the clarity of this sentence by focusing only on the size reduction.
>
> > l331: Has $b$ been introduced?
>
> Thank you for pointing this out. We write $b$ to denote the bit complexity of the input matrix $\mathbf A$, that is, each entry of $\mathbf A$ can be represented by $b$ bits. This is stated in Theorem D.1 of our initial draft, and we will move this definition to the main text in our revision.
>
> > Lemma 3.1: Please check the statement. Are the order of quantifiers correct? Should the bound on $s$ be moved further back?
>
> Thanks for bringing this to our attention. The bound on $s$ does not need to be moved, since the set $\mathcal{S}$ only depends on $n$, $k$, and $\epsilon$. However, it should say: ``for every $\mathbf{A}$ and $\mathbf{B}$ there exists $\mathbf{S} \in \mathcal{S}$...''. We will fix this.
>
> > I think $S$ is $m\times n$. Same for l380.
>
> Thank you for catching this, we have fixed this in our revision.

---

### Official Review · Reviewer_ZwUH · 2023-07-26

**Soundness:** 3 good
**Presentation:** 3 good
**Contribution:** 3 good
**Rating:** 5
**Confidence:** 4

**Summary:**

This paper presents results for the k-means and sparse dictionary problems, both of which ask to summarize an $n$ point data set in $d$ dimensions in terms of $k$ points. In the former we map each point to a center, in the latter, we are allowed sparse linear combinations of points. The paper considers two models, the streaming model (various versions of it) and the "standard" model where the goal is to comes up with an algrithm that is polynomial in $n. d$,  but $k, \epsilon$ are treated as constants, and the dependence on these can be arbitrary.  They present both upper and lower bounds.

Their lower bound results are:
- An $\Omega(n/\epsilon)$ streaming lower bound for $k$ means clustering. This beats the trivial $\Omega(n)$ lower bound but it falls short of the $O(n/\epsilon^2)$ upper bound from JL. The proof is by a reduction from set disjointness, as is standard in streaming. The authors argue that their reduction is delicate and uses the structure of the hard instances from BYJKS'04.
- An $\Omega(dk/\epsilon)$ lower bound which follows from earlier work by Woodruff.
- They give some other lower bounds for restricted models.

They also give results on PTASES for both problems. The idea behind both is to reduce the dimensionality of the points using various kinds of sketches. The exact sketches needed for these are chosen with some care. In low dimensions, one can afford a brute-force enumeration or similarly costly algorithm (this general idea goes back to the early work on coresets). They also give some results in the turnstile streaming model, but the results seem to have some caveats about the parameters/solution space.








**Strengths:**

- The problems considered are important and well-studied in the literature, the results will be of interest to people working in the general area os sketching/streaming.

- I like the fact that they give unified results for $k$-means and the sparse dictionary problems.

- The results seem to rely on a deep understanding of the prior work in the area, and on using exactly the right tools needed in each setting.

**Weaknesses:**

- The paper has too many results, at least some of them rather partial or for rather restricted models. I have a hard time deciding what the main contribution of the paper is. No one result stood out either in terms of the statement, or in terms of new techniques.

- Some of the results seem a touch incremental, they come from applying prior ideas in a new setting. I realize that knowing what tools are applicable is no mean feat, given the vast literature. But I could not discern too much originality.

**Questions:**

If you wanted to reader to focus on one result or key idea which you see as the main contribution of your work, what would it be?

I would suggest that the writeup focus on one or two main results, and defer the other results for the expert reader.

---

> ### Author Rebuttal · Authors · 2023-08-10
>
> > The paper has too many results, at least some of them rather partial or for rather restricted models. I have a hard time deciding what the main contribution of the paper is. No one result stood out either in terms of the statement, or in terms of new techniques.
>
> > I would suggest that the writeup focus on one or two main results, and defer the other results for the expert reader.
>
> Because the main contribution of our work is an exploration of a problem/setting which has not received much attention in the past, we believe that this work would be incomplete without including discussions about some of our more straightforward or partial results. We have attempted to highlight two technical highlights (discussed further in the next response) in Section 1.1. We will try to rewrite this section, as well as subsequent discussions, to focus more on these technical highlights, while still mentioning our other results and their connections to the landscape of results for PTAS and streaming algorithms for sparse dictionary learning and k-means clustering.
>
> > If you wanted the reader to focus on one result or key idea which you see as the main contribution of your work, what would it be?
>
> The overall message is that we initiate the study of k-means clustering and sparse dictionary learning (with assignment) in the PTAS and turnstile streaming settings, which we show are both amenable to techniques from the sketching literature. We would also like to reiterate that, surprisingly, prior work has not considered the task of outputting assignments for $k$-means or sparse dictionary learning in a stream despite its practical importance, and we hope our work will bring much-needed attention to this important problem. Technically, perhaps the two most interesting results (requiring the most "technical novelty") we would like to highlight are the first PTAS for sparse dictionary learning, as well as an $\Omega(n/\epsilon)$ lower bound for k-means clustering in turnstile streams. The first result uses a new reduction to coresets for projective clustering, while the latter uses a reduction to multi-party set disjointness (rather than the standard two-party problem), which requires delicate arguments to reason about optimal solutions to random instances of k-means clustering.
>
> > Some of the results seem a touch incremental, they come from applying prior ideas in a new setting. I realize that knowing what tools are applicable is no mean feat, given the vast literature. But I could not discern too much originality.
>
> While our techniques indeed rely on several standard results for our main technical results, such as the use of coresets for the sparse dictionary learning PTAS and set disjointness lower bounds for the k-means clustering communication lower bound, we argue that several innovations are still needed to make our proofs go through.
>
> For our sparse dictionary learning PTAS, the use of coresets to design a PTAS is indeed a well-known idea. However, the standard idea of building coresets for this problem by first computing an approximately optimal solution does not work in our setting, as the computation of an approximate solution is our goal to begin with. Instead, we show a reduction to coresets for projective clustering, which is a new connection to the best of our knowledge, and furthermore use the fact that there exist constructions for such coresets that do not need an approximately optimal solution, which uses a recent work of Tukan et al 2022. Note that older coreset constructions for projective clustering (e.g. https://people.csail.mit.edu/dannyf/stoc11.pdf) do not have this property.
>
> For our k-means clustering communication lower bound, one of the main challenges we face is to understand the optimal cost of a k-means clustering instance up to a $(1+\epsilon)$ relative factor, which is highly nontrivial since k-means clustering is an NP-hard problem in general, and we must exploit the structure of the specific instance at hand in order to characterize the cost of instances. This challenge is further complicated by the fact that our instance must be a dense random instance in order to use the set disjointness lower bounds, unlike some other lower bound results in other models shown in earlier work which have simpler instances to reason about that are supported on standard basis vectors (e.g., https://arxiv.org/abs/1905.06394, https://arxiv.org/abs/2202.12793). We introduce new techniques such as boosting the "signal" of the clustering instance with only a small decrease in the communication lower bound by using multi-party set disjointness, which we believe is a new interesting application of multi-party lower bounds. Note that with a standard two-party set disjointness argument, a nearly optimal k-means clustering may not necessarily solve the set disjointness problem, since the cost savings from clustering this coordinate correctly is only two, while the cost savings from finding the optimal clustering of the random bits could be much larger.
>
> Additionally, our turnstile streaming algorithms relies on a delicate argument combining a sketched multiple linear regression guarantee, affine-embedding, and JL-embedding. This combination must be done carefully to balance the space of the sketch with the strength of the guarantee needed, which we do by conceptually breaking the problem into disjoint pieces of 1) reducing the problem using the linear regression guarantee, 2) solving the reduced problem with the affine embedding, and 3) checking if we have solved the overall problem with a JL-embedding guarantee. Achieving this decomposition relies on leveraging a less well-known ``guess-the-sketch'' approach.

---

### Decision · Program_Chairs · 2023-09-21

**Decision:**

Accept (poster)

**Comment:**

The paper has general support from all reviewers.

On the plus side, the paper considers two important problems of k-means and sparse dictionary learning in a natural setting where the assignments of the input points are required in addition to the centers/dictionary. The paper has a lot of results, both upper and lower bounds drawing from a lot of techniques in sketching.

On the minus side, the reviewers find the techniques somewhat standard to the area and the work seems to be about finding the right combinations of existing techniques. That said, the literature in this area is vast and it is a nontrivial task to identify the right techniques. Several reviewers find the paper lacking a major result that stands out in terms of technique or statement but rather having a large number of good results. The reviewers recognized that the main contribution of the paper was theoretical and focused on the strength of the theory. In addition and to a lesser extent, the practicality of the algorithms is also questioned by several reviewers due to the lack of experiments and the very high (exponential in k with terms such as k^(k^r) or n^(k^2)) theoretical runtime.

The authors are strongly advised to improve the writing of the paper by highlighting the key contributions and adding the explanations from the discussion with the reviewers.